# Proximity-Informed Calibration for Deep Neural Networks

**Miao Xiong**[1][*]    **Ailin Deng**[1]    **Pang Wei Koh**[23]    **Jiaying Wu**[1]
**Shen Li**[1]    **Jianqing Xu**    **Bryan Hooi**[1]
[1] National University of Singapore [2] University of Washington [3] Google

## Abstract

Confidence calibration is central to providing accurate and interpretable uncertainty estimates, especially under safety-critical scenarios. However, we find that existing calibration algorithms often overlook the issue of *proximity bias*, a phenomenon where models tend to be more overconfident in low proximity data (i.e., data lying in the sparse region of the data distribution) compared to high proximity samples, and thus suffer from inconsistent miscalibration across different proximity samples. We examine the problem over $504$ pretrained ImageNet models and observe that: 1) Proximity bias exists across a wide variety of model architectures and sizes; 2) Transformer-based models are relatively more susceptible to proximity bias than CNN-based models; 3) Proximity bias persists even after performing popular calibration algorithms like temperature scaling; 4) Models tend to overfit more heavily on low proximity samples than on high proximity samples. Motivated by the empirical findings, we propose PROCAL, a plug-and-play algorithm with a theoretical guarantee to adjust sample confidence based on proximity. To further quantify the effectiveness of calibration algorithms in mitigating proximity bias, we introduce proximity-informed expected calibration error (PIECE) with theoretical analysis. We show that PROCAL is effective in addressing proximity bias and improving calibration on balanced, long-tail, and distribution-shift settings under four metrics over various model architectures. We believe our findings on proximity bias will guide the development of *fairer and better-calibrated* models, contributing to the broader pursuit of trustworthy AI.

## 1 Introduction

Machine learning systems are increasingly deployed in high-stakes applications such as medical diagnosis [30, 36, 9, 6], where incorrect decisions can have severe human health consequences. To ensure safe and reliable deployment, *confidence calibration* approaches [10, 23, 28] are employed to produce more accurate uncertainty estimates, which allow models to establish trust by communicating their level of uncertainty, and to defer to human decision-making when the models are uncertain.

In this paper, we present a calibration-related phenomenon termed *proximity bias*, which refers to the tendency of current deep classifiers to exhibit higher levels of overconfidence on samples of low proximity, i.e., samples in sparse areas within the data distribution (see Figure 1 for an illustrative example). In this study, we quantify the proximity of a sample (Eq. 1) using the average distance to its $K$ (e.g. $K = 10$) nearest neighbor samples in the data distribution, and we observe that proximity bias holds for various choices of $K$. Importantly, the phenomenon persists even after applying existing popular calibration methods, leading to different levels of miscalibration across proximities.

The proximity bias issue raises safety concerns in real-world applications, particularly for underrepresented populations (i.e. low proximity samples) [30, 35]. A recent skin cancer analysis highlights

---

[*]Miao Xiong (miao.xiong@u.nus.edu).

37th Conference on Neural Information Processing Systems (NeurIPS 2023).

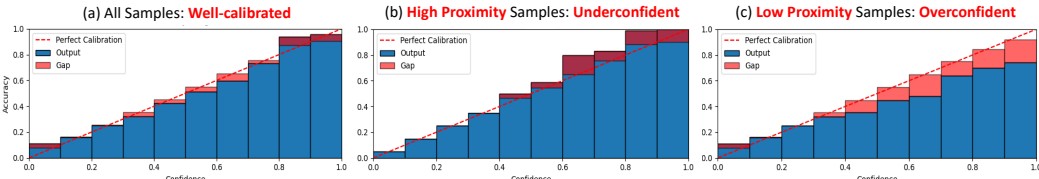

Figure 1: **Samples with lower (higher) proximity tend to be more overconfident (underconfident)**. The results are conducted using XCiT, an Image Transformer, on the ImageNet validation set (**All Samples**). The sample's proximity is measured using the average distance to its nearest neighbors ($K = 10$) in the validation set. We split samples into 10 equal-size bins based on proximity and choose the bin with the highest proximity (**High Proximity Samples**) and lowest proximity (**Low Proximity Samples**).

this concern by revealing that AI-powered models demonstrate high performance for light-skinned individuals but struggle with dark-skinned individuals due to their underrepresentation [11]. This issue can also manifest in the form of proximity bias: suppose a dark-skinned individual has a high risk of 40% of having the cancer. However, due to their underrepresentation within the data distribution, the model *overconfidently* assigns them 98% confidence of not having cancer. As a result, these low proximity individuals may be deprived of timely intervention.

To study the ubiquity of this problem, we examine $504$ ImageNet pretrained models from the `timm` library [43] and make the following key observations: 1) Proximity bias exists generally across a wide variety of model architectures and sizes; 2) Transformer-based models are relatively more susceptible to proximity bias than CNN-based models; 3) Proximity bias persists even after performing popular calibration algorithms including temperature scaling; 4) Low proximity samples are more prone to model overfitting while high proximity samples are less susceptible to this issue.

Besides, we argue that proximity bias is overlooked by *confidence calibration*. Revisiting its definition, $\mathbb{P}(Y = \hat{Y} \mid \hat{P} = p) = p$ for all $p \in [0, 1]$, we find that its primary goal is to match confidence with the accuracy of samples sharing the same confidence level. However, Figure 1a reveals that although the model seems well-calibrated within each confidence group, there still exists miscalibration errors among these groups (e.g. low and high proximity samples) due to proximity bias.

Motivated by this, we propose a debiased variant of the expected calibration error (ECE) metric, called proximity-informed expected calibration error (PIECE) to further capture the miscalibration error due to proximity bias. The effectiveness is supported by our theoretical analysis that PIECE is at least as large as ECE and this equality holds when there is no cancellation effect with respect to proximity bias.

To tackle proximity bias and further improve confidence calibration, we propose a plug-and-play method, PROCAL. Intuitively, PROCAL learns a joint distribution of proximity and confidence to adjust probability estimates. To fully leverage the characteristics of the input information, we develop two separate algorithms tailored for continuous and discrete inputs. We evaluate the algorithms on large-scale datasets: **balanced datasets** including ImageNet [7] and Yahoo-Topics [49], **long-tail datasets** iNaturalist 2021 [3] and ImageNet-LT [27] and **distribution-shift datasets** MultiNLI [44] and ImageNet-C [15]. The results show that our algorithm consistently improves the performance of existing algorithms under four metrics with 90% significance (p-value < 0.1).

Our main contributions can be summarized as follows:

- **Findings**: We discover the proximity bias issue and show its prevalence over large-scale analysis (504 ImageNet pretrained models).
- **Metrics**: To quantify the effectiveness of mitigating proximity bias, we introduce proximity-informed expected calibration error (PIECE) with theoretical analysis.
- **Method Effectiveness**: We propose a plug-and-play method PROCAL with theoretical guarantee and verify its effectiveness on various image and text settings.

## 2 Related Work

**Confidence Calibration** Confidence calibration aims to yield uncertainty estimates via aligning a model's confidence with the accuracy of samples with the same confidence level [10, 25, 28].

To achieve this, **Scaling-based** methods, such as temperature scaling [10], adjust the predicted probabilities by learning a temperature scalar for all samples. Similarly, parameterized temperature scaling [39] offers improved expressiveness via input-dependent temperature parameterization, and Mix-n-Match [48] adopts ensemble and composition strategies to yield data-efficient and accuracy-preserving estimates. **Binning-based** methods divide samples into multiple bins based on confidence and calibrate each bin. Popular methods include classic histogram binning [46], mutual-information-maximization-based binning [32], and isotonic regression [47]. However, existing calibration methods overlook the proximity bias issue, which fundamentally limits the methods' capabilities in delivering reliable and interpretable uncertainty estimates.

**Multicalibration**   Multicalibration algorithms [13, 21] aim to achieve a certain level of fairness by ensuring that a predictor is well-calibrated for the overall population as well as different computationally-identifiable subgroups. [33] proposes a grouping loss to evaluate subgroup calibration error while we propose a metric to integrate the group cancellation effect into existing calibration loss. [21] focuses on understanding the fundamental trade-offs between group calibration and other fairness criteria, and [13] proposes a conceptual iterative algorithm to learn a multi-calibrated predictor. In this regard, our proposed framework can be considered a specific implementation of the fairness objectives outlined in [13], with a particular focus on proximity-based subgroups. This approach offers easier interpretation and implementation compared to subgroups discussed in [13, 21].

## 3   What is Proximity Bias?

In this section, we study the following questions: What is proximity bias? When and why does proximity bias occur?

**Background**   We consider a supervised multi-class classification problem, where input $X \in \mathcal{X}$ and its label $Y \in \mathcal{Y} = \{1, 2, \cdots, C\}$ follow a joint distribution $\pi(X, Y)$. Let $f$ be a classifier with $f(X) = (\hat{Y}, \hat{P})$, where $\hat{Y}$ represents the predicted label, and $\hat{P}$ is the model's confidence, i.e. the estimate of the probability of correctness [10]. For simplicity, we use $\hat{P}$ to denote both the model's confidence and the confidence calibrated using existing calibration algorithms.

**Proximity**   We define proximity as a function of the average distance between a sample $X$ and its $K$ nearest neighbors $\mathcal{N}_K(X)$ in the data distribution:

$$D(X) = \exp\left(-\frac{1}{K} \sum_{X_i \in \mathcal{N}_K(X)} \text{dist}(X, X_i)\right), \tag{1}$$

where $\text{dist}(X, X_i)$ denotes the distance between sample $X$ and its $i$-th nearest neighbor $X_i$, estimated using Euclidean distance between the features of $X$ and $X_i$ from the model's penultimate layer. $K$ is a hyperparameter (we set $K = 10$ in this paper). We use the validation set as a proxy to estimate the data distribution. That is, we compute any point's proximity by finding its nearest neighbors in the held-out validation set. Although the training set can also be employed to compute proximity, we utilize the validation set because it is readily accessible during the calibration process.

The exponential function is used to normalize the distance measure from a range of $[0, \inf]$ to $[0, 1]$, making the approach more robust to the effects of distance scaling since the absolute distance in Euclidean distance can cause instability and difficulty in modeling. This definition allows us to capture the local density of a sample and its relationship to its neighborhood. For instance, a sample situated in a sparse region of the training distribution would receive a low proximity value, while a sample located in a dense region would receive a high proximity value. Samples with low proximity values represent **underrepresented samples** in the data distribution that merit attention, such as rare ("long-tail") diseases, minority populations, and samples with distribution shift.

**Proximity Bias**   To investigate the relationship between proximity and model miscalibration, we define proximity bias as follows:

**Definition 3.1.** Given any confidence level $p$, the model suffers from proximity bias if the following condition does not hold:

$$\mathbb{P}\left(\hat{Y} = Y \mid \hat{P} = p, D = d_1\right) = \mathbb{P}\left(\hat{Y} = Y \mid \hat{P} = p, D = d_2\right) \quad \forall d_1, d_2 \in (0, 1], d_1 \neq d_2.$$

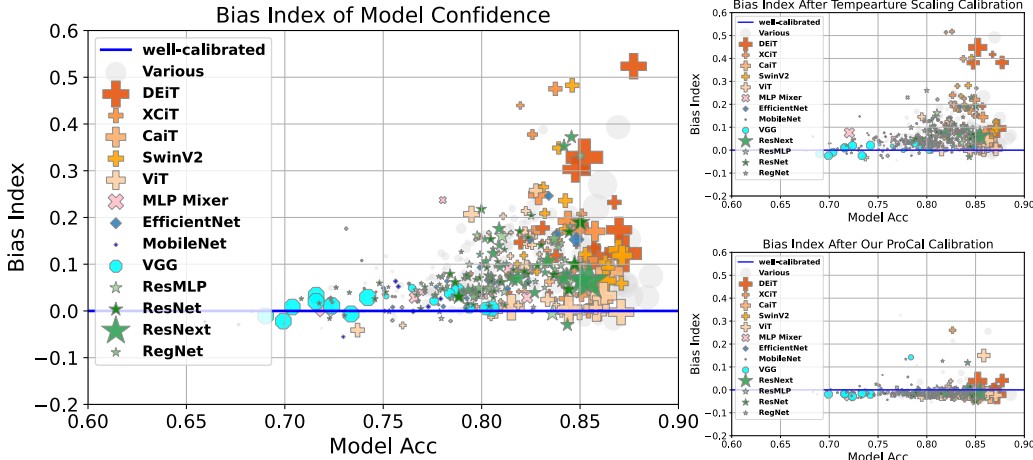

Figure 2: Proximity bias analysis on $504$ public models. Each marker represents a model, where marker sizes indicate model parameter numbers and different colors/shapes represent different architectures. The bias index is computed using Equation (3) (0 indicates no proximity bias). **Left**: We observed the following: 1) Models with higher accuracy tend to have a larger bias index. 2) Proximity bias exists across a wide range of model architectures. 3) Transformer variants (e.g. DEiT, XCiT, CaiT, and SwinV2) have a relatively larger bias compared to convolution-based networks (e.g. VGG and ResNet variants). **Right**: Confidence calibrated by temperature scaling (Upper Right) is similar to the original model confidence w.r.t proximity bias. Our PROCAL (Bottom Right) is effective in reducing proximity bias. Analysis of other existing calibration algorithms can be found in Appendix D.

The intuition behind this definition is that, ideally, a sample with a confidence level of $p$ should have a probability of being correct equal to $p$, regardless of proximity. However, if low proximity samples consistently display higher confidence than high proximity samples (as shown in Figure 1), it can lead to unreliable and unjust decision-making, particularly for underrepresented populations.

## 3.1 Main Empirical Findings

To showcase the ubiquity of this problem, we examine the proximity bias phenomenon on $504$ ImageNet pretrained models from the `timm` library [43] and show the results in Figure 2 (see Appendix D for additional figures and analysis). We use statistical hypothesis testing to investigate the presence of proximity bias. The null hypothesis $H_0$ is that proximity bias does not exist, formally, for any confidence $p$ and proximities $d_1 > d_2$:

$$\mathbb{P}\left(\hat{Y} = Y \mid \hat{P} = p, D = d_1\right) = \mathbb{P}\left(\hat{Y} = Y \mid \hat{P} = p, D = d_2\right). \quad (2)$$

To test the above null hypothesis, we first split the samples into 5 equal-sized proximity groups and select the highest and lowest proximity groups. From the high proximity group, we randomly select 10,000 points and find corresponding points in the low proximity group that have similar confidence levels. Next, we reverse this process, randomly selecting 10,000 points from the low proximity group and find corresponding points in the high proximity group with matched confidence. We then merge all the points from the high proximity group into $B_H$ and those from the low proximity group into $B_L$, with the $B_H$ and $B_L$ having similar average confidence. Finally, we apply the Wilcoxon rank-sum test [24] to evaluate whether there is a significant difference in the sample means (i.e. accuracy) of $B_H$ and $B_L$. More implementation details can be found in Appendix C.

Inspired by the hypothesis testing, we define **Bias Index** as the accuracy drop between the confidence-matched high proximity group $B_H$ and low proximity group $B_L$ to reflect the degree of bias:

$$\text{Bias Index} = \frac{\sum_{(X,Y) \in B_H} \mathbb{1}\{\hat{Y} = Y\}}{|B_H|} - \frac{\sum_{(X,Y) \in B_L} \mathbb{1}\{\hat{Y} = Y\}}{|B_L|} = \text{Acc}(B_H) - \text{Acc}(B_L). \quad (3)$$

Note that $B_H, B_L$ are obtained from the hypothesis testing process and hence have the same mean confidence.

The hypothesis testing results indicate that over 80% of $504$ models have a p-value less than $0.05$ (72% after Bonferroni correction [4]), i.e., the null hypothesis is rejected with a confidence level of at least 95%, indicating that proximity bias plagues most of the models in `timm`.

We show the bias index of $504$ models in Figure 2 and make the following findings:

**1. Proximity bias exists generally across a wide variety of model architecture and sizes.** Figure 2 shows that most models (80% of the models as supported by hypothesis testing) have a bias index larger than 0, indicating the existence of proximity bias.

**2. Transformer-based methods are relatively more susceptible to proximity bias than CNN-based methods.** In Figure 2, models with lower accuracy (primarily CNN-based models such as VGG, EfficientNet, MobileNet, and ResNet [12] variants) tend to have lower bias index. On the other hand, among models with higher accuracy, Transformer variants (e.g., DEiT, XCiT [1], CaiT [40], and SwinV2) demonstrate relatively higher bias compared to convolution-based networks (e.g., ResNet variants). This is concerning given the increasing popularity of Transformer-based models in recent years and highlights the need for further research to study and address this issue.

**3. Popular calibration methods such as temperature scaling do not noticeably alleviate proximity bias.** Figure 2 (upper right) shows that the proximity bias index remains large even after applying temperature scaling, indicating that this method does not noticeably alleviate the problem. In contrast, Figure 2c demonstrates that our proposed approach successfully shifts the models to a much closer distribution around the line $y = 0$ (indicating no proximity bias). The bias index figures for more existing calibration methods are provided in Appendix D.

**4. Low proximity samples are more prone to model overfitting.** Figure 4 in Appendix D shows that the model's accuracy difference between the training and validation set is more significant on low proximity samples (31.67%) compared to high proximity samples (0.6%). This indicates that the model generalizes well on samples of high proximity but tends to overfit on samples of low proximity. The overconfidence of low proximity samples can be a consequence of the overfitting tendency, as the overfitting gap also reflects the mismatch between the model's confidence and its actual accuracy.

## 4 Proximity-Informed ECE

As depicted in Figure 1, *existing evaluation metrics underestimate the true miscalibration level*, as proximity bias causes certain errors in the model to cancel out. As an example, consider a scenario:

**Example 4.1.** All samples are only drawn from two proximity groups of equal probability mass, $d = 0.2$ and $d = 0.8$, with true probabilities of $\mathbb{P}(Y = \hat{Y} | X, f)$ being $0.5$ and $0.9$, respectively. The model outputs the confidence score $p = 0.7$ to all samples.

We consider the most commonly used metric, expected calibration error (ECE) that is defined as $\text{ECE} = \mathbb{E}_{\hat{P}}\left[\left|\mathbb{P}(\hat{Y} = Y \mid \hat{P}) - \hat{P}\right|\right]$. In Example 4.1, the ECE is 0, suggesting that the model is perfectly calibrated in terms of confidence calibration. In fact, the model has significant miscalibration issues: it is heavily overconfident in one proximity group while heavily underconfident in the other, highlighting the limitations of existing calibration metrics. The miscalibration errors within the same confidence group are *canceled out* by samples with both high and low proximity, resulting in a phenomenon we term *cancellation effect*.

To further evaluate the miscalibration canceled out by proximity bias, we propose the proximity-informed expected calibration error (PIECE). PIECE is defined in an analogous fashion as ECE, yet it further examines information about the proximity of the input sample, $D(X)$, in the calibration evaluation:

$$\text{PIECE} = \mathbb{E}_{\hat{P}, D}\left[\left|\mathbb{P}(\hat{Y} = Y \mid \hat{P}, D) - \hat{P}\right|\right]. \tag{4}$$

Back to Example 4.1 where ECE = 0, we have PIECE = 0.2, revealing its miscalibration level in the subpopulations of different proximities, i.e., the calibration error regarding proximity bias. Additionally, we demonstrate in Theorem 4.2 that PIECE is *always at least as large* as ECE, with the equality holding only when there is no cancellation effect w.r.t proximity. The detailed proof is relegated to Appendix B.

**Theorem 4.2** (PIECE captures cancellation effect.). *Given any joint distribution $\pi(X, Y)$ and any classifier $f$ that outputs model confidence $\hat{P}$ for sample $X$, we have the following inequality, where equality holds only when there is no cancellation effect with respect to proximity:*

$$\underbrace{\mathbb{E}_{\hat{P}}\left[\left|\mathbb{P}(\hat{Y}=Y \mid \hat{P})-\hat{P}\right|\right]}_{\text{ECE}} \leq \underbrace{\mathbb{E}_{\hat{P},D}\left[\left|\mathbb{P}(\hat{Y}=Y \mid \hat{P},D)-\hat{P}\right|\right]}_{\text{PIECE}}.$$

## 5 How to Mitigate Proximity Bias?

In this section, we propose PROCAL to achieve three goals: 1) **mitigate proximity bias**, i.e., ensure samples with the same confidence level have the same miscalibration gap across all proximity levels, 2) **improve confidence calibration** by reducing overconfidence and underconfidence, and 3) provide a **plug-and-play** method that can combine the strengths of existing approaches with our proximity-informed approach.

The high-level intuition is to explicitly incorporate proximity when estimating the underlying probability of the model prediction being correct. In addition, existing calibration algorithms can be classified into 2 types: 1) those producing *continuous outputs*, exemplified by scaling-based methods [10]; 2) those producing *discrete outputs*, such as binning-based methods, which group the samples into bins and assign the same scores to samples within the same bin. To fully leverage the distinct properties of the input information, we develop two separate algorithms tailored for continuous and discrete inputs. This differentiation is based on the observation that continuous outputs (e.g., those produced by scaling-based methods) contain rich distributional information suitable for density estimation. On the other hand, discrete inputs (e.g., those generated by binning-based methods) allow for robust binning-based adjustments. By treating these inputs separately, we can effectively harness the characteristics of each type.

In summary, Density-Ratio Calibration (§5.1) estimates continuous density functions, and aligns well with type 1) methods that produce continuous confidence scores. In contrast, Bin-Mean-Shift (§5.2) does not rely on densities, making it more compatible with type 2) calibration methods that yield discrete outputs. Together, these two calibration techniques constitute a versatile plug-and-play framework PROCAL, applicable to confidence scores of both continuous and discrete types.

### 5.1 Continuous Confidence: Density-Ratio Calibration

The common interpretation of confidence is the likelihood of a model prediction $\hat{Y}$ being identical to the ground truth label $Y$ for every sample $X$. Computing this probability directly with density estimation methods can be computationally demanding, particularly in high-dimensional spaces. To circumvent the curse of dimensionality and address proximity bias, we incorporate the model confidence $\hat{P}$ and proximity information $D(X)$ to estimate the posterior probability of correctness, i.e., $\mathbb{P}(\hat{Y}=Y \mid \hat{P}, D)$. This approach is data-efficient since it conducts density estimation in a two-dimensional space only, rather than in the higher dimensional feature or prediction simplex space.

Consider a test sample $X$ with proximity $D = D(X)$ and uncalibrated confidence score $\hat{P}$, which can be the standard Maximum Softmax Probability (MSP), or the output of any calibration method. $\mathbb{P}(\hat{Y}=Y \mid \hat{P}, D)$ can be computed via Bayes' rule:

$$\mathbb{P}\left(\hat{Y}=Y \mid \hat{P}, D\right) = \frac{\mathbb{P}\left(\hat{P}, D \mid \hat{Y}=Y\right) \mathbb{P}\left(\hat{Y}=Y\right)}{\mathbb{P}(\hat{P}, D)},$$

where $\hat{Y}$ is the model prediction and $Y$ is the ground truth label. This can be re-expressed as follows by using the law of total probability:

$$\frac{\mathbb{P}(\hat{P}, D \mid \hat{Y}=Y)}{\mathbb{P}\left(\hat{P}, D \mid \hat{Y}=Y\right) + \mathbb{P}\left(\hat{P}, D \mid \hat{Y} \neq Y\right) \cdot \frac{\mathbb{P}(\hat{Y} \neq Y)}{\mathbb{P}(\hat{Y}=Y)}}.$$

To compute this calibrated score, we need to estimate the distributions $\mathbb{P}\left(\hat{P}, D \mid \hat{Y} = Y\right)$ and $\mathbb{P}\left(\hat{P}, D \mid \hat{Y} \neq Y\right)$, and the class ratio $\frac{\mathbb{P}(\hat{Y} \neq Y)}{\mathbb{P}(\hat{Y} = Y)}$.

To estimate the probability density functions $\mathbb{P}\left(\hat{P}, D \mid \hat{Y} = Y\right)$ and $\mathbb{P}\left(\hat{P}, D \mid \hat{Y} \neq Y\right)$, various density estimation methods can be used, such as parametric methods like Gaussian mixture models or non-parametric methods like kernel density estimation (KDE)[31]. We choose KDE because it is flexible and robust, making no assumptions about the underlying distribution (see Appendix C for specific implementation details). Specifically, we split samples into two groups based on whether they are correctly classified and then use KDE to estimate the two densities. To obtain the class ratio $\frac{\mathbb{P}(\hat{Y} \neq Y)}{\mathbb{P}(\hat{Y} = Y)}$, we simply use the ratio of the number of correctly classified samples and the number of misclassified samples in the validation set. The **pseudocode** for inference and training can be found in Appendix 2 and 1.

## 5.2 Discrete Confidence: Bin Mean-Shift

The Bin-Mean-Shift approach aims to first use 2-dimensional binning to estimate the joint distribution of proximity $D$ and input confidence $\hat{P}$ and then estimate $\mathbb{P}\left(\hat{Y} = Y \mid \hat{P}, D\right)$. Considering test samples $X$ with proximity $D = D(X)$ and uncalibrated confidence score $\hat{P}$, we first group samples into 2-dimensional equal-size bins based on their $D$ and $\hat{P}$ (other binning schemes can also be used; we choose quantile for simplicity). Next, for each bin $B_{mh}$, we calculate its accuracy $\mathcal{A}(B_{mh})$ and mean confidence $\mathcal{F}(B_{mh})$. Then, the confidence scores of samples within the bin are adjusted as:

$$\hat{P}_{ours} = \hat{P} + \lambda \cdot \left(\mathcal{A}(B_{mh}) - \mathcal{F}(B_{mh})\right), \tag{5}$$

where the shrinkage coefficient $\lambda \in (0, 1]$ is a hyper-parameter, controlling the bias-variance trade-off. Ideally, setting $\lambda = 1$ would ideally achieve our goal. However, in practice, we often encounter bins with a smaller number of samples, whose estimate of $\mathcal{A}(B_{mh}) - \mathcal{F}(B_{mh})$ will have high variance and therefore inaccurate. To reduce variance in these scenarios, we can set a smaller $\lambda$. In practice, we choose $\lambda = 0.5$ as a reasonable default for all our experiments, which we find offers consistent performance across various settings.

Note that our approach (i.e. applying a mean-shift in each bin) differs from the typical histogram binning method (replacing the confidence scores with its mean accuracy in each bin). Rather than completely replacing the input confidence scores $\hat{P}$ (which are often reasonably well-calibrated), our approach better utilizes these scores by only adjusting them by the minimal mean-shift needed to correct for proximity bias in each of the 2-dimensional bins.

## 5.3 Theoretical Guarantee

Here we present that our method, Bin-Mean-Shift, can consistently achieve a smaller Brier Score given a sufficient amount of data in the context of binary classification. The Brier Score [5] is a strictly proper score function that measures both calibration and accuracy aspects [22], with a smaller value indicating better performance. As illustrated below, our algorithm's Brier Score is asymptotically bounded by the original Brier Score, augmented by a non-negative term.

**Theorem 5.1** (Brier Score after Bin-Mean-Shift is asymptotically bounded by Brier Score before calibration)**.** *Given a joint data distribution $\pi(X, Y)$ and a binary classifier $f$, for any calibration algorithm $h$ that outputs score $h(\hat{P})$ based on model confidence $\hat{P}$, we apply Bin-Mean-Shift to derive calibrated score $\tilde{h}(\hat{P})$ as defined in Equation* (5)*. Let $h_c(\hat{P}) = h(\hat{P}) \times \mathbb{1}\{\hat{Y} = 1\} + (1 - h(\hat{P})) \times \mathbb{1}\{\hat{Y} = 0\}$ denote the probability assigned to class 1 by $h(\hat{P})$, and define $\tilde{h}_c(\hat{P})$ similarly. Then, the Brier Score before calibration can be decomposed as follows:*

$$\underbrace{\mathbb{E}_{\pi(X,Y)}\left[\left(h_c(\hat{P}) - Y\right)^2\right]}_{\text{Brier Score before Calibration}} = \underbrace{\mathbb{E}_{\pi(X,Y)}\left[\left(\tilde{h}_c(\hat{P}) - Y\right)^2\right]}_{\text{Brier Score after Calibration}} + \underbrace{\mathbb{E}_{B \sim \mathbb{P}(B)}\left[\left(\hat{\mathcal{A}}(B) - \hat{\mathcal{F}}(B)\right)^2\right]}_{\geq 0} + o(1),$$

*where $\mathbb{P}(B)$ is determined by the binning mechanism used in Bin-Mean-Shift.*

Table 1: Calibration performance of ImageNet pretrained ResNet50 on long-tail dataset iNaturalist 2021. * denotes significant improvement (p-value < 0.1). 'Base' refers to existing calibration methods, 'Ours' to our method applied to calibration. Note that 'Conf+Ours' shows the result of our method applied directly to model confidence. Calibration error is given by $\times 10^{-2}$.

| Method | ECE ↓ | | ACE ↓ | | MCE ↓ | | PIECE ↓ | |
| | base | +ours | base | +ours | base | +ours | base | +ours |
|---|---|---|---|---|---|---|---|---|
| Conf | 4.85 | **0.78**\* | 4.86 | **0.76**\* | 0.55 | **0.18**\* | 4.91 | **1.51**\* |
| TS | 2.03 | **0.70**\* | 2.02 | **0.78**\* | 0.30 | **0.14**\* | 2.34 | **1.43**\* |
| ETS | 1.12 | **0.66**\* | 1.15 | **0.77**\* | 0.18 | **0.13**\* | 1.79 | **1.38**\* |
| PTS | 4.86 | **0.71**\* | 4.90 | **0.86**\* | 2.96 | **0.12** | 7.04 | **1.44**\* |
| PTSK | 2.97 | **0.65**\* | 3.01 | **0.82**\* | 0.56 | **0.11**\* | 4.66 | **1.41**\* |
| MIR | 1.05 | **0.98** | 1.09 | **1.06** | **0.18** | 0.19 | **1.63** | 1.64 |

**Remark.** Note that when the calibration algorithm $h$ is an identity mapping, it demonstrates that Bin-Mean-Shift achieves better model calibration performance than the original model confidence, given a sufficient amount of data. The detailed proof is relegated to Appendix A.

# 6  Experiments

In this section, we aim to answer the following questions:

- **Performance across different datasets and model architectures:** How does PROCAL perform on datasets with balanced distribution, long-tail distribution, and distribution shift, as well as on different model architectures?
- **Inference efficiency:** How efficient is our PROCAL? (see Appendix E.1)
- **Hyperparameter sensitivity**: How sensitive is PROCAL to different hyperparameters, e.g. neighbor size $K$? (See Appendix F.2)
- **Ablation study:** What is the difference between Density-Ratio and Bin-Mean-shift on calibration? How should the choice between these techniques be determined? (See Appendix F.1)

## 6.1  Experiment Setup

**Evaluation Metrics.** Following [10], we adopt 3 commonly used metrics to evaluate the *confidence calibration*: Expected Calibration Error (ECE), Adaptive Calibration Error (ACE [29]), Maximum Calibration Error (MCE) and our proposed PIECE to evaluate the *bias mitigation* performance. More detailed introduction of these metrics can be found in Appendix C.

**Datasets.** We evaluate the effectiveness of our approach across large-scale datasets of three types of data characteristics (balanced, long-tail and distribution-shifted) in image and text domains: (1) Dataset with **balanced** class distribution (i.e. each class has an equal size of samples) on vision dataset ImageNet [7] and two text datasets including Yahoo Answers Topics [49] and MultiNLI-match [44]; (2) Datasets with **long-tail** class distribution on two image datasets, including iNaturalist 2021 [3] and ImageNet-LT [27]; (3) Dataset with **distribution-shift** on three datasets, including ImageNet-C [15], MultiNLI-Mismatch [44] and ImageNet-Sketch [42].

**Comparison methods.** We compare our method to existing calibration algorithms: base confidence score (Conf) [16], *scaling-based methods* such as Temperature Scaling (TS) [10], Ensemble Temperature Scaling (ETS) [48], Parameterized Temperature Scaling (PTS) [39], Parameterized Temperature Scaling with K Nearest Neighbors (PTSK), and *binning based methods* such as Histogram Binning (HB), Isotonic Regression (IR) and Multi-Isotonic Regression (MIR) [48]. Throughout the experiment section, we apply Density-Ratio Calibration to Conf, TS, ETS, PTS, and PTSK and apply Bin-Mean-Shift to binning-based methods IR, HB, and MIR. HB and IR are removed from the long-tail setting due to its instability when the class sample size is very small.

More details on baseline algorithms, datasets, pretrained models, hyperparameters, and implementation details can be found in Appendix C.

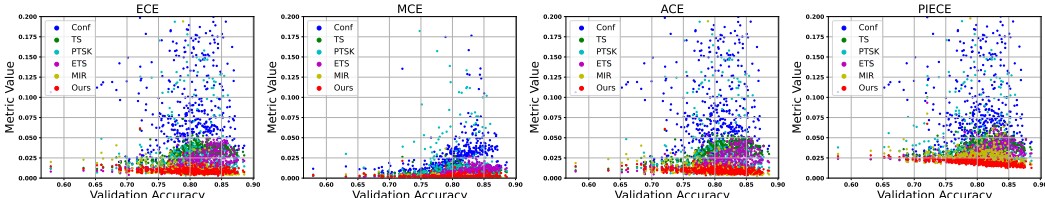

Figure 3: Calibration errors on ImageNet across 504 `timm` models. Each point represents the calibration result of applying a calibration method to the model confidence. Marker colors indicate different calibration algorithms used. Among all calibration algorithms, our method consistently appears at the bottom of the plot. See Appendix E Figure 11 for high resolution figures.

Table 2: We use RoBERTa models [26] fine-tuned on Yahoo and MultiNLI Match, respectively, as their models. 'Base' refers to existing calibration methods and 'Ours' refers to our method applied to existing calibration methods. Calibration error is given by $\times 10^{-2}$.

| Method | ECE ↓ | | ACE ↓ | | MCE ↓ | | PIECE ↓ | | Method | ECE ↓ | | ACE ↓ | | MCE ↓ | | PIECE ↓ | |
|---|---|---|---|---|---|---|---|---|---|---|---|---|---|---|---|---|---|
| | base | +ours | base | +ours | base | +ours | base | +ours | | base | +ours | base | +ours | base | +ours | base | +ours |
| Conf | 4.56 | **0.51** | 4.56 | **0.54** | 0.84 | **0.09** | 4.73 | **1.32** | Conf | 2.47 | **1.45** | 2.62 | **1.46** | 0.85 | **0.37** | 3.54 | **2.78** |
| TS | 0.50 | **0.42** | 0.46 | 0.46 | 0.09 | **0.07** | 2.22 | **1.34** | TS | 1.70 | **1.22** | 1.76 | **1.35** | 0.41 | **0.40** | 3.03 | **2.78** |
| ETS | 0.62 | **0.43** | 0.59 | **0.45** | 0.09 | **0.07** | 2.22 | **1.37** | ETS | 1.58 | **1.24** | 1.56 | **1.26** | 0.66 | **0.39** | 3.04 | **2.77** |
| PTS | 0.55 | **0.42** | 0.52 | **0.42** | 0.10 | **0.09** | 2.03 | **1.41** | PTS | 7.53 | **2.42** | 7.50 | **2.46** | 4.21 | **0.93** | 7.78 | **3.56** |
| PTSK | 0.61 | **0.47** | 0.51 | 0.51 | 0.13 | **0.09** | 2.15 | **1.38** | PTSK | 10.14 | **4.26** | 10.14 | **4.40** | 7.40 | **2.83** | 10.33 | **5.15** |
| HB | 3.28 | **1.99** | 4.88 | **2.09** | 1.68 | **0.67** | 5.33 | **2.92** | HB | 1.11 | **1.05** | **1.17** | 1.50 | 0.36 | **0.25** | 3.76 | **2.51** |
| IR | **0.64** | 0.84 | **0.64** | 0.77 | **0.13** | 0.19 | 2.08 | **1.75** | IR | **0.88** | 1.43 | **1.07** | 1.23 | 0.36 | **0.34** | 2.46 | 2.61 |
| MIR | **0.63** | 0.72 | **0.54** | 0.61 | **0.11** | 0.18 | 2.08 | **1.65** | MIR | **0.71** | 1.03 | **1.07** | 1.36 | **0.26** | 0.40 | 2.58 | **2.35** |

(a) Yahoo Answer Topics

(b) MultiNLI Mismatch

## 6.2 Effectiveness

**Datasets with the balanced class distribution.** The results on ImageNet of 504 models from `timm` [43] are depicted in Figure 3, where our method (red color markers) consistently appears at the bottom, achieving the lowest calibration error across all four evaluation metrics in general. This indicates that our method consistently outperforms other approaches in eliminating proximity bias and improving confidence calibration. We also select four popular models from these 504 models, specifically `BeiT` [2], `MLP Mixer` [38], `ResNet50` [12] and `ViT` [8]. A summary of their results is presented in Table 4 of Appendix E. Additionally, Table 2a and Table 5 present the calibration results for the text classification task on Yahoo Answers Topics and the text understanding task on MultiNLI, where our method consistently improves the calibration of existing methods and model confidence. SeeAppendix E for more details.

**Datasets with the long-tail class distribution.** Table 1 shows the results on the long-tail image dataset iNaturalist 2021. Our method ('ours') consistently improves upon existing algorithms ('base') regarding reducing confidence calibration errors (`ECE`, `ACE`, and `MCE`) and mitigating proximity bias (`PIECE`). Note that even when used independently ('Conf+ours') without combining with existing algorithms, our method achieves the best performance across all metrics. This result suggests that our algorithm can make the model more calibrated in the long-tail setting by effectively mitigating the bias towards low proximity samples (i.e. tail classes), highlighting its practicality in real-world scenarios where data is often imbalanced and long-tailed. ImageNet-LT results in Table 6 of Appendix E.3 show similar improvement.

**Datasets with distribution shift.** Table 2b shows our method's calibration performance when trained on an in-distribution validation set (MultiNLI Match) and applied to a cross-domain test set (MultiNLI Mismatch). The results suggest that our method can improve upon most existing methods on ECE, ACE and MCE, and gain consistent improvement on PIECE, indicating its effectiveness in mitigating proximity bias. Moreover, empirical results on ImageNet-C (Figure 12) and ImageNet-Sketch (Table 7) also demonstrate consistent improvements of our method over baselines. Besides, compared to Bin-Mean-Shift, Density-Ratio exhibits more stable performance on enhancing the existing baselines. More analysis on their comparison can be found in Appendix F.1.

# 7 Conclusions and Discussion

In this paper, we focus on the problem of proximity bias in model calibration, a phenomenon wherein deep models tend to be more overconfident on data of low proximity (i.e. lying in the sparse region of data distribution) and thus suffer from miscalibration. We study this phenomenon on $504$ public models across a wide variety of model architectures and sizes on ImageNet and find that the bias persists even after applying the existing calibration methods, which drives us to propose PROCAL for tackling proximity bias. To further evaluate the miscalibration due to proximity bias, we propose a proximity-informed expected calibration error (PIECE) with theoretical analysis. Extensive empirical studies on balanced, long-tail, and distribution-shifted datasets under four metrics support our findings and showcase the effectiveness of our method. [2]

**Potential Impact, Limitations and Future Work**   We uncover the proximity bias phenomenon and show its prevalence through large-scale analysis, highlighting its negative impact on the safe deployment of deep models, e.g. unfair decisions on minority populations and false diagnoses for underrepresented patients. We also provide PROCAL as a starting point to mitigate the proximity bias, which we believe has the potential to inspire more subsequent works, serve as a useful guidance in the literature and ultimately lead to *improved and fairer decision-making in real-world applications*, especially for underrepresented populations and safety-critical scenarios. However, our study also has several limitations. First, our PROCAL maintains a held-out validation set during inference for computing proximity. While we have shown that the cost can be marginal for large models (see Inference Efficiency in Appendix E.1), it may be challenging if applied to small devices where memory is limited. Future research can investigate the underlying mechanisms of proximity bias and explore various options to replace the existing approach of local density estimation. Additionally, we only focus on the closed-set multi-class classification problem; future work can generalize this to multi-label, open-set or generative settings.

## Acknowledgments

The author extends gratitude to Yao Shu and Zhongxiang Dai for the insightful discussions during the review response period, and to Yifei Li for providing constructive feedback on the manuscript draft. This research is supported by the National Research Foundation Singapore under its AI Singapore Programme (Award Number: [AISG2-TC-2021-002]).

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

# A Proof of Theorem 5.1

**Notation.** Let $\pi(X, Y)$ denote the true underlying distribution of the sample $X$ and its label $Y$. The empirical dataset used for Bin-Mean-Shift is denoted by $D_n = \{(\mathbf{x}_1, y_1), (\mathbf{x}_2, y_2), \ldots, (\mathbf{x}_n, y_n)\}$, where $n$ is the number of samples in the dataset, and each datapoint $(\mathbf{x}_i, y_i)$ is independently and identically sampled from $\pi(X, Y)$.

A classifier $f$ is defined as $f(X) = (\hat{Y}, \hat{P})$, where $\hat{Y}$ represents the predicted label and $\hat{P}$ represents the model's confidence, i.e., the estimate of the probability of correctness [10].

We use $h(\hat{P})$ to denote the output confidence score for class prediction $\hat{Y}$ calibrated by any calibration algorithm $h$. For example, in the simplest case, $h$ can be the identity mapping, in which case $h(\hat{P})$ represents the original model confidence.

To simplify notation, we use $B = B(X)$ to denote the bucket to which $X$ belongs. Given a bucket $B$, we compute the empirical accuracy $\hat{\mathcal{A}}_n(B)$ and mean confidence score $\hat{\mathcal{F}}_n(B)$ for samples in the bucket, based on the empirical dataset $D_n$. Given the classifier $f$ and calibrator $h$, they are computed as follows:

$$\hat{\mathcal{A}}_n(B) = \frac{1}{|B|} \sum_{(\mathbf{x}, y) \in D_n} \mathbb{1}\{(y = \hat{y}) \wedge (B(\mathbf{x}) = B)\} \tag{6}$$

$$\hat{\mathcal{F}}_n(B) = \frac{1}{|B|} \sum_{(\mathbf{x}, y) \in D_n \wedge (\mathbf{x} \in B)} h(\hat{P}) \tag{7}$$

Additionally, we use $\mathcal{A}(B)$ and $\mathcal{F}(B)$ to denote the expected confidence and actual accuracy of samples from the underlying data distribution $\pi(X, Y)$ and belonging to the bucket $B$:

$$\mathcal{A}(B) = \mathbb{E}_{\pi(X, Y)}\left[\mathbb{1}\{Y = \hat{Y}\} \mid B(X) = B\right] \tag{8}$$

$$\mathcal{F}(B) = \mathbb{E}_X\left[h(\hat{P}) \mid B(X) = B\right] \tag{9}$$

With these definitions, our proposed Bin-Mean-Shift algorithm can be expressed as:

$$\tilde{h}(\hat{P}) = h(\hat{P}) + \hat{\mathcal{A}}_n(B) - \hat{\mathcal{F}}_n(B), \tag{10}$$

where $B$ is the bucket that the input sample $X$ belongs to, and $\tilde{h}(\hat{P})$ is the score calibrated using our Bin-Mean-Shift algorithm.

First we revisit the definition of Brier Score. The Brier Score [5] is a strictly proper score function that measures both calibration and accuracy aspects [22], with a smaller value indicating better performance. The Brier Score is defined as mean square loss as follows for binary classification:

$$\text{Brier Score} = \mathbb{E}_{\pi(X, Y)}\left[\left(h_c(\hat{P}) - Y\right)^2\right], \tag{11}$$

where $h_c(\hat{P})$ is the probability assigned to class 1 by the calibration algorithm $h$:

$$h_c(\hat{P}) = \begin{cases} h(\hat{P}) & \text{when } \hat{Y} = 1 \\ 1 - h(\hat{P}) & \text{when } \hat{Y} = 0 \end{cases} \tag{12}$$

In short, this can be represented as $h_c(\hat{P}) = h(\hat{P}) \times \mathbb{1}\{\hat{Y} = 1\} + (1 - h(\hat{P})) \times \mathbb{1}\{\hat{Y} = 0\}$.

**Theorem A.1** (Brier Score after Bin-Mean-Shift is asymptotically bounded by Brier Score before calibration)**.** *Given a joint data distribution $\pi(X, Y)$ and a binary classifier $f$, for any calibration algorithm $h$ that outputs a score $h(\hat{P})$ based on model confidence $\hat{P}$, we apply Bin-Mean-Shift to derive the calibrated score $\tilde{h}(\hat{P})$ as defined in Equation (10). Let $h_c(\hat{P}) = h(\hat{P}) \times \mathbb{1}\{\hat{Y} = 1\} + (1 - h(\hat{P})) \times \mathbb{1}\{\hat{Y} = 0\}$ denote the probability assigned to class 1 by $h(\hat{P})$, and define $\tilde{h}_c(\hat{P})$ similarly. Then, the Brier Score before calibration can be decomposed as follows:*

$$\underbrace{\mathbb{E}_{\pi(X, Y)}\left[\left(h_c(\hat{P}) - Y\right)^2\right]}_{\textit{Brier Score before Calibration}} = \underbrace{\mathbb{E}_{\pi(X, Y)}\left[\left(\tilde{h}_c(\hat{P}) - Y\right)^2\right]}_{\textit{Brier Score after Calibration}} + \underbrace{\mathbb{E}_{B \sim \mathbb{P}(B)}\left[\left(\hat{\mathcal{A}}_n(B) - \hat{\mathcal{F}}_n(B)\right)^2\right]}_{\geq 0} + o(1),$$

*where $\mathbb{P}(B)$ is determined by the binning mechanism used in Bin-Mean-Shift.*

*Proof.* First we prove the following equality which re-expresses the Brier score in an equivalent form:

$$\mathbb{E}_{\pi(X,Y)}\left[\left(h_c(\hat{P}) - Y\right)^2\right] = \mathbb{E}_{\pi(X,Y)}\left[\left(h(\hat{P}) - \tilde{Y}\right)^2\right], \tag{13}$$

where $\tilde{Y} = \mathbb{1}\{Y = \hat{Y}\}$ is a binary variable representing whether the model's prediction is correct:

$$\tilde{Y} = \mathbb{1}\{Y = \hat{Y}\} = \begin{cases} \mathbb{1}\{Y = 0\} = 1 - Y, & \text{when} \quad \hat{Y} = 0 \\ \mathbb{1}\{Y = 1\} = Y, & \text{when} \quad \hat{Y} = 1 \end{cases} \tag{14}$$

Then we consider $h_c(\hat{P}) - Y$. By Eq. (12):

$$(h_c(\hat{P}) - Y)^2 = \begin{cases} (1 - h(\hat{P}) - Y)^2, & \text{when} \quad \hat{Y} = 0 \\ (h(\hat{P}) - Y)^2, & \text{when} \quad \hat{Y} = 1 \end{cases} \tag{15}$$

$$= \begin{cases} (\tilde{Y} - h(\hat{P}))^2, & \text{when} \quad \hat{Y} = 0 \\ (h(\hat{P}) - Y)^2, & \text{when} \quad \hat{Y} = 1 \end{cases} \tag{16}$$

So we have $(h_c(\hat{P}) - Y)^2 = (h(\hat{P}) - \tilde{Y})^2$ which completes the proof of the equality in Eq. (13).

Using this, we rewrite the Brier score as follows:

$$\text{Brier Score} = \mathbb{E}_{\pi(X,Y)}\left[\left(h(\hat{P}) - \tilde{Y}\right)^2\right], \tag{17}$$

Second, we decompose the Brier score:

$$\mathbb{E}_{\pi(X,Y)}\left[\left(h(\hat{P}) - \tilde{Y}\right)^2\right] = \mathbb{E}_{\pi(X,Y)}\left[\left(h(\hat{P}) - \tilde{h}(\hat{P}) + \tilde{h}(\hat{P}) - \tilde{Y}\right)^2\right] \tag{18}$$

$$= \underbrace{\mathbb{E}_{\pi(X,Y)}\left[\left(\tilde{h}(\hat{P}) - \tilde{Y}\right)^2\right]}_{(a)} \tag{19}$$

$$+ \underbrace{\mathbb{E}_{\pi(X,Y)}\left[\left(h(\hat{P}) - \tilde{h}(\hat{P})\right)^2\right]}_{(b)} \tag{20}$$

$$+ \underbrace{2\mathbb{E}_{\pi(X,Y)}\left[\left(h(\hat{P}) - \tilde{h}(\hat{P})\right)\left(\tilde{h}(\hat{P}) - \tilde{Y}\right)\right]}_{(c)} \tag{21}$$

First note that term $(a)$ is the Brier score after calibration (recalling our earlier equivalent form for the Brier score). For term $(b)$ we recall that $\tilde{h}(\hat{P}) = h(\hat{P}) + \hat{\mathcal{A}}_n(B) - \hat{\mathcal{F}}_n(B)$, which is our proposed Bin-Mean-Shift algorithm in Equation (10). Note that $X$ can be sampled by first sampling the bin $B$ and then sampling the point $X$ from the corresponding bin. So term $(b)$ can be expressed as:

$$\mathbb{E}_{\pi(X,Y)}\left[\left(h(\hat{P}) - \tilde{h}(\hat{P})\right)^2\right] = \mathbb{E}_{\pi(X,Y)}\left[\left(h(\hat{P}) - h(\hat{P}) - \hat{\mathcal{A}}_n(B) + \hat{\mathcal{F}}_n(B))\right)^2\right]$$

$$= \mathbb{E}_{\pi(X,Y)}\left[\left(\hat{\mathcal{A}}_n(B) - \hat{\mathcal{F}}_n(B))\right)^2\right]$$

$$= \mathbb{E}_{B \sim \mathbb{P}(B)}\mathbb{E}_{X \sim \mathbb{P}(X|B)}\left[\left(\hat{\mathcal{A}}_n(B(X)) - \hat{\mathcal{F}}_n(B(X))\right)^2\right]$$

$$= \mathbb{E}_{B \sim \mathbb{P}(B)}\left[\left(\hat{\mathcal{A}}_n(B) - \hat{\mathcal{F}}(B)\right)^2\right]$$

where $\hat{\mathcal{A}}_n(B(X))$ and $\hat{\mathcal{F}}_n(B(X))$ remain the same for all samples following into the same bucket $B$.

For term $(c)$ we show that:

$$\mathbb{E}_{\pi(X,Y)}\left[\left(h(\hat{P}) - \tilde{h}(\hat{P})\right)\left(\tilde{h}(\hat{P}) - \tilde{Y}\right)\right] \tag{22}$$

$$= \mathbb{E}_{\pi(X,Y)}\left[\left(\hat{\mathcal{F}}(B(X)) - \hat{\mathcal{A}}(B(X))\right)\left(\tilde{h}(\hat{P}) - \tilde{Y}\right)\right] \tag{23}$$

$$= \mathbb{E}_{B \sim \mathbb{P}(B)}\mathbb{E}_{(X,Y) \sim \mathbb{P}(X,Y|B)}\left[\left(\hat{\mathcal{F}}(B(X)) - \hat{\mathcal{A}}(B(X))\right)\left(\tilde{h}(\hat{P}) - \tilde{Y}\right)\right] \tag{24}$$

$$= \mathbb{E}_{B \sim \mathbb{P}(B)}\left[\left(\hat{\mathcal{A}}_n(B) - \hat{\mathcal{F}}_n(B)\right)\underbrace{\mathbb{E}_{(X,Y) \sim \mathbb{P}(X,Y|B)}\left[\tilde{Y} - \tilde{h}(\hat{P})\right]}_{(d)}\right], \tag{25}$$

where in the last step $(\hat{\mathcal{A}}_n(B) - \hat{\mathcal{F}}_n(B))$ is a function of $B$ and thus can be moved out of the inner expectation. For term $(d)$ we have:

$$\mathbb{E}_{(X,Y) \sim \mathbb{P}(X,Y|B)}\left[\tilde{Y} - \tilde{h}(\hat{P})\right] \tag{26}$$

$$= \mathbb{E}_{(X,Y) \sim \mathbb{P}(X,Y|B)}\left[\tilde{Y} - h(\hat{P}) - \hat{\mathcal{A}}_n(B) + \hat{\mathcal{F}}_n(B)\right] \tag{27}$$

$$= \mathbb{E}_{(X,Y) \sim \mathbb{P}(X,Y|B)}\left[\left(\tilde{Y} - \hat{\mathcal{A}}_n(B)\right) + \left(\hat{\mathcal{F}}_n(B) - h(\hat{P})\right)\right] \tag{28}$$

$$= \underbrace{\left(\mathbb{E}_{(X,Y) \sim \mathbb{P}(X,Y|B)}\left[\tilde{Y}\right] - \hat{\mathcal{A}}_n(B)\right)}_{(e)} + \left(\hat{\mathcal{F}}_n(B) - \mathcal{F}(B)\right) \tag{29}$$

For term $(e)$ we have:

$$\mathbb{E}_{(X,Y) \sim \mathbb{P}(X,Y|B)}\left[\tilde{Y}\right] = \mathbb{E}_{(X,Y) \sim \mathbb{P}(X,Y|B)}\left[\mathbb{1}\{Y = \hat{Y}\}\right] = \mathcal{A}(B). \tag{30}$$

By the Law of Large Numbers, the sample means converges to their expectations as follows:

$$\lim_{n \to \infty} \hat{\mathcal{F}}_n(B) = \mathcal{F}(B) \quad \text{and} \quad \lim_{n \to \infty} \hat{\mathcal{A}}_n(B) = \mathcal{A}(B). \tag{31}$$

This further leads to the following statement for term $(c)$:

$$\lim_{n \to \infty} \mathbb{E}_{\pi(X,Y)}\left[\left(h(\hat{P}) - \tilde{h}(\hat{P})\right)\left(\tilde{h}(\hat{P}) - \tilde{Y}\right)\right] = 0. \tag{32}$$

Then finally we have the statement:

$$\lim_{n \to \infty} \underbrace{\mathbb{E}_{\pi(X,Y)}\left[\left(h(\hat{P}) - Y\right)^2\right]}_{\textbf{Brier Score before Calibration}} - \underbrace{\mathbb{E}_{\pi(X,Y)}\left[\left(\tilde{h}_c(\hat{P}) - Y\right)^2\right]}_{\textbf{Brier Score after Calibration}} - \underbrace{\mathbb{E}_{B \sim \mathbb{P}(B)}\left[\left(\hat{\mathcal{A}}_n(B) - \hat{\mathcal{F}}_n(B)\right)^2\right]}_{\geq 0} = 0$$

$\square$

**Discussion** This theorem shows that Bin-Mean-Shift (BMS) preserves the calibration properties of the input scores $\hat{P}$; i.e. if $h(\hat{P})$ was already well-calibrated, the BMS-calibrated scores $\tilde{h}(\hat{P})$ will continue to be well-calibrated (up to an $o(1)$ term). At the same time, BMS helps to correct for miscalibrations with respect to any choice of buckets (represented by the $(\hat{\mathcal{A}}_n(B) - \hat{\mathcal{F}}_n(B))^2$ term).

Interestingly, this theorem implies that we can have theoretical guarantees for a *pipeline* of different calibration methods: for example, consider a pipeline consisting of any calibration method $h(\hat{P})$, followed by one or more applications of BMS with different choices of binning schemes. Then this theorem shows that the Brier score will decrease with each application of BMS (setting aside the $o(1)$ term). Thus, in contrast to the theoretical guarantees of existing calibration methods (such as histogram binning), which only apply to a single calibration method, our approach points to the theoretical benefits of such pipelines of calibration methods.

# B    Proof of PIECE Guarantee

Recall that $f$ is a classifier (e.g. neural network) with $f(X) = (\hat{Y}, \hat{P})$, where $\hat{Y}$ represents the predicted label, and $\hat{P}$ is the model's confidence, i.e. the estimate of the probability of correctness [10].

**Theorem B.1** (PIECE captures cancellation effect.). *Given any joint distribution $\pi(X, Y)$ and any classifier $f$ that outputs model confidence $\hat{P}$ for sample $X$, we have the following inequality, where equality holds only when there is no cancellation effect with respect to proximity:*

$$\underbrace{\mathbb{E}_{\hat{P}}\left[\left|\mathbb{P}(\hat{Y} = Y \mid \hat{P}) - \hat{P}\right|\right]}_{\text{ECE}} \leq \underbrace{\mathbb{E}_{\hat{P}, D}\left[\left|\mathbb{P}(\hat{Y} = Y \mid \hat{P}, D) - \hat{P}\right|\right]}_{\text{PIECE}}.$$

*Proof.* Note that ECE [10] is defined as:

$$\text{ECE} = \mathbb{E}_{\hat{P}}\left[\left|\mathbb{P}(\hat{Y} = Y \mid \hat{P}) - \hat{P}\right|\right] \tag{33}$$

We first consider the formula within the expectation:

$$
\begin{aligned}
\left|\mathbb{P}(\hat{Y} = Y \mid \hat{P}) - \hat{P}\right| &= \left|\mathbb{E}_{X,Y}\left[\mathbb{I}\{\hat{Y} = Y\} \mid \hat{P}\right] - \hat{P}\right| \\
&= \left|\mathbb{E}_{X,Y}\left[\mathbb{I}\{\hat{Y} = Y\} - \hat{P} \mid \hat{P}\right]\right| \\
&\overset{(a)}{=} \left|\mathbb{E}_D\left[\mathbb{E}_{X,Y}\left[\mathbb{I}\{\hat{Y} = Y\} - \hat{P} \mid \hat{P}, D\right]\right]\right| \\
&\overset{(b)}{\leq} \mathbb{E}_D\left[\left|\mathbb{E}_{X,Y}\left[\mathbb{I}\{\hat{Y} = Y\} - \hat{P} \mid \hat{P}, D\right]\right|\right] \\
&= \mathbb{E}_D\left[\left|\mathbb{E}_{X,Y}\left[\mathbb{I}\{\hat{Y} = Y\} \mid \hat{P}, D\right] - \hat{P}\right|\right] \\
&= \mathbb{E}_D\left[\left|\mathbb{P}(\hat{Y} = Y \mid \hat{P}, D) - \hat{P}\right|\right]
\end{aligned}
\tag{34}
$$

where $(a)$ is derived using the Law of Total Expectation and $(b)$ is derived using Jensen's Inequality.

The equality holds if and only if $\left|\mathbb{P}(\hat{Y} = Y \mid \hat{P}, D) - \hat{P}\right|$ is a linear function in terms of $D$. This condition is satisfied only when there is no cancellation effect with respect to proximity $D$, i.e. either $\mathbb{P}(\hat{Y} = Y \mid \hat{P}, D) \geq \hat{P}$ or $\mathbb{P}(\hat{Y} = Y \mid \hat{P}, D) \leq \hat{P}$ hold for all choices of $D$.

Then applying this formula to Equation 33, we have:

$$
\begin{aligned}
\text{ECE} &= \mathbb{E}_{\hat{P}}\left[\left|\mathbb{P}(\hat{Y} = Y \mid \hat{P}) - \hat{P}\right|\right] \\
&\leq \mathbb{E}_{\hat{P}}\left[\mathbb{E}_D\left[\left|\mathbb{P}(\hat{Y} = Y \mid \hat{P}, D) - \hat{P}\right|\right]\right] \\
&= \mathbb{E}_{\hat{P}, D}\left[\left|\mathbb{P}(\hat{Y} = Y \mid \hat{P}, D) - \hat{P}\right|\right] \\
&= \text{PIECE}
\end{aligned}
\tag{35}
$$

# C    Experimental Setup

**Evaluation Metrics.**    Following [10, 29], we adopt three commonly used metrics to evaluate the *confidence calibration*: Expected Calibration Error (ECE), Adaptive Calibration Error (ACE [29]), Maximum Calibration Error (MCE) and our proposed PIECE to evaluate the *bias mitigation* performance. To compute these metrics, we first divide samples into $M = 15$ bins and compute every bin's average confidence and accuracy. Then we compute the absolute difference between each bin's average confidence and its corresponding accuracy. The final calibration error is measured using the weighted difference (the fraction of samples in each bin as the weight). The key distinction between ECE and ACE lies in the binning scheme: ECE divides bins with *equal-confidence* intervals while ACE uses an adaptive scheme that spaces the bin intervals to contain an *equal number* of samples in each bin. In addition, PIECE splits the bin both based on confidence and proximity. It splits samples into $M = 15$ equal-number bins based confidence and then split every confidence group into $H = 10$ equal-number bins based on proximity. MCE chooses the bin with the largest difference between average confidence and accuracy and output their absolute difference as the final error.

**Datasets.**    We evaluate the effectiveness of our approach across large-scale datasets of three types of data characteristics (balanced, long-tail and distribution-shifted) in image and text domains: (1) Dataset with **balanced** class distribution (i.e. each class has an equal size of samples) on vision dataset ImageNet [7] and two text datasets including Yahoo Answers Topics [49] and MultiNLI-match [44]; (2) Datasets with **long-tail** class distribution on two image datasets, including iNaturalist 2021 [3] and ImageNet-LT [27]; (3) Dataset with **distribution-shift** on three datasets, including ImageNet-C [15], MultiNLI-Mismatch [44] and ImageNet-Sketch [42].

**Comparison methods.**    We compare our method to existing calibration algorithms: base confidence score (`Conf`) [16], *scaling-based methods* such as Temperature Scaling (`TS`) [10], Ensemble Temperature Scaling (`ETS`) [48], Parameterized Temperature Scaling (`PTS`) [39], Parameterized Temperature Scaling with K Nearest Neighbors (`PTSK`), and *binning based methods* such as Histogram Binning (`HB`), Isotonic Regression (`IR`) and Multi-Isotonic Regression (`MIR`) [48]. Throughout the experiment section, we apply Density-Ratio Calibration to Conf, TS, ETS, PTS, and PTSK and apply Bin-Mean-Shift to binning-based methods IR, HB, and MIR. HB and IR are removed from the long-tail setting due to its instability when the class sample size is very small.

**Pretrained Models.**    For the proximity bias analysis in section 3 and the balanced ImageNet evaluation, we use 504 pretrained models from timm [43] (the list of models are shown in the code repository). For ImageNet-LT evaluation, we use the model ResNext50 pretrained by Liu et al. [27] using classifier re-training technique. For iNaturalist 2021, we directly use the ImageNet pretrained backbone ResNet [12] which we follow this paper [41] and download from the repo[3].. For Yahoo Answers Topics and MultiNLI datasets, we use pretrained RoBERTa from HuggingFace API and fine-tuned on the corresponding datasets with 3 epochs.

**Hyperparamerters.**    Regarding nearest neighbor computation, we use indexFlatL2 from faiss [18]. Except the Hyperparamerter sensitivity experiments, we use $K = 10$ for the proximity computation. Regarding our method, for Density-Ratio, the kernel density estimation for two variables are implemented using statsmodel library [37]. For the Bin Mean-Shift method, we set the regularization parameter $\lambda = 0.5$. For the calibration setup, we adopt a standard calibration setup [10] with a fixed-size calibration set (i.e. validation set) and evaluation test datasets. Specifically, we randomly split the hold-out dataset into calibration set and evaluation set $n_c = n_e = 25000$ for ImageNet, $n_c = n_e = 50000$ for iNaturalist 2021 and $n_c = n_e = 5000$ for ImageNet-LT. For Yahoo and MultiNLI-Match dataset, we sample 20% data from the training dataset as calibration set and use the original test dataset as the test dataset. For the evaluation, we use random seed 2020, 2021, 2022, 2023, 2024 and compute the mean (this does not apply to NLP dataset for its fixed test set).

**Details on Statistical Hypothesis Testing.**    The statistical hypothesis testing requires that for two samples with the same confidence but different proximity, their probability of being correct is the same. To achieve this, we aim for every pair of high and low proximity samples to have the same confidence levels. However, there is indeed an observable difference in the average confidence of $B_H$ and $B_L$, with high proximity samples $B_H$ having higher average confidence. This leads to certain samples that do not have a counterpart with equivalent confidence in the other proximity group. To address this issue, we employ a nearest neighbor search combined with a rejection policy. Specifically, pairs with a confidence difference greater than 0.05 are discarded. This ensures that we are only pairing samples from different proximity groups that have the closest possible confidence levels to one another. When implementing this algorithm, users can also visualize the confidence distribution to verify whether the confidence of two proximity groups have evident overlapping. If their confidence levels have no overlap, we suggest reducing the number of splits from 5 to 3 to ensure low/high proximity groups have similar confidence but different proximity.

**Details on KDE**    For its simplicity and effectiveness, we use the *KDEMultivariate* function from the *statsmodel* library for density estimation. This function employs a Gaussian Kernel and applies the normal reference rule of thumb (i.e. bw=$1.06\hat{\sigma}n^{-\frac{1}{5}}$) based on the the standard deviation $\hat{\sigma}$ and sample size $n$ to select an appropriate bandwidth. While it is possible to use other density estimation kernels such as *Exponential Kernel* in *Scikit Learn*, we found that the Gaussian kernel coupled with

---

[3]https://github.com/visipedia/newt/blob/main/benchmark/README.mdv

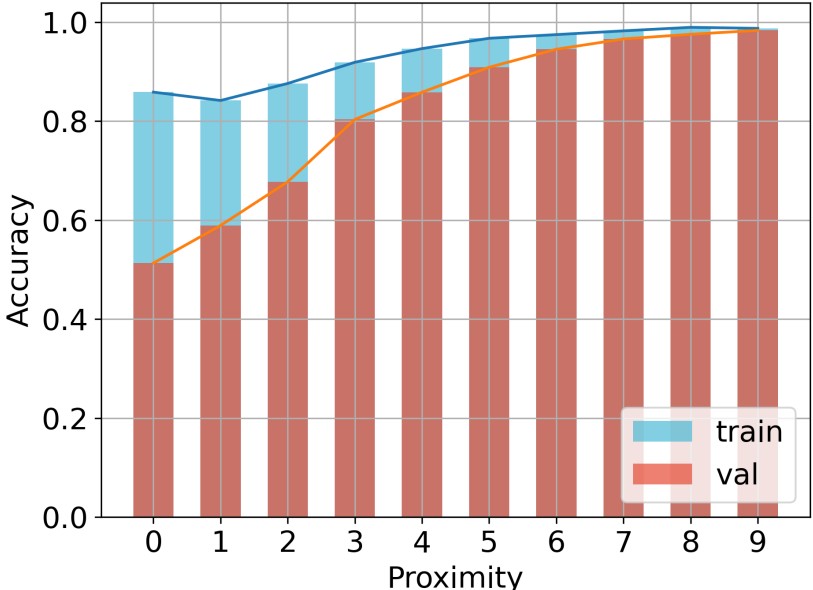

Figure 4: The model's accuracy difference between the training and validation set is more significant on low proximity samples (31.67%) compared to high proximity samples (0.6%). The discrepancy in accuracy between the training and validation sets increases as the samples approach to low proximity regions, despite the training dataset and validation set have overlapping proximity distributions.

the normal reference rule for bandwidth selection generally yields better performance across various models and datasets.

□

## D  Additional Empirical Findings

### D.1  Low proximity samples are more prone to model overfitting

To study the behavior of low proximity samples and high proximity samples, we compute their accuracy difference between training dataset and validation set. Figure 4 reveals a clear tendency: the model's accuracy difference between the training and validation set is more significant on low proximity samples (31.67%) compared to high proximity samples (0.6%). This indicates that the model generalizes well on samples of high proximity but tends to overfit on samples of low proximity. The overconfidence of low proximity samples can be a consequence of the overfitting tendency, as the overfitting gap also reflects the mismatch between the model's confidence and its actual accuracy.

### D.2  Proximity Bias Across A Variety of Models

Here we present the bias index of several popular calibration algorithms on 504 models on ImageNet. As depicted in Figure 5, most models have a bias index larger than 0, indicating the existence of proximity bias across a variety of models. Notably, even after applying temperature scaling, as demonstrated in Figure 6, the calibrated model confidences still exhibit proximity bias. This tendency is further corroborated by Figure 7 (after multi isotonic regression calibration) and Figure 8 (after ensemble temperature scaling). In stark contrast, Figure 9 reveals the effectiveness of our proposed approach in mitigating proximity bias.

### D.3  Confidence and Accuracy Are Positively Correlated With Proximity

Our initial investigation delves into the relationship between sample proximity, confidence, and accuracy across a variety of deep neural network models. We observe a clear trend wherein the

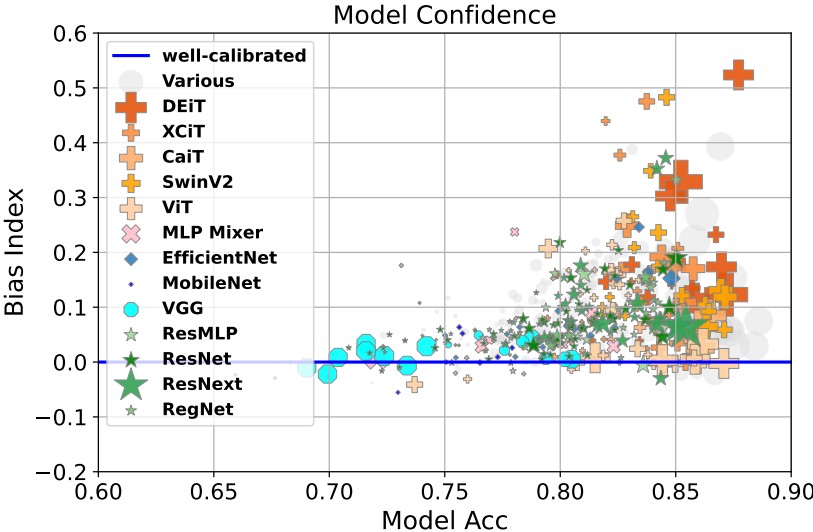

Figure 5: Proximity bias analysis of the model confidence on $504$ public models. Each marker represents a model, where marker sizes indicate model parameter numbers and different colors/shapes represent different architectures. The bias index is computed using Equation (3) (0 indicates no proximity bias).

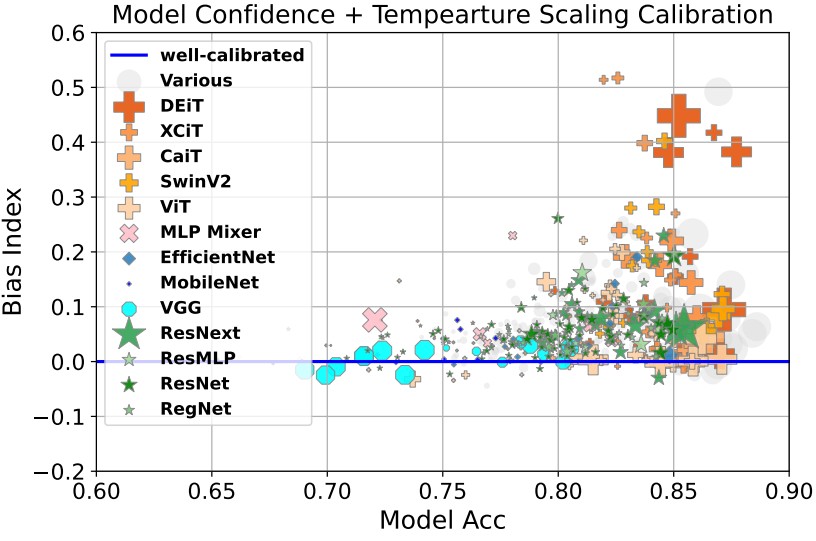

Figure 6: Proximity bias analysis of the model confidence calibrated using **temperature scaling** on $504$ public models. Each marker represents a model, where marker sizes indicate model parameter numbers and different colors/shapes represent different architectures. The bias index is computed using Equation (3) (0 indicates no proximity bias).

model's confidence and accuracy exhibit an upward trajectory from low proximity samples to high proximity samples. This trend is illustrated in Figure 10.

This tendency suggests that the model is less confident with samples from low proximity regions, where training samples are sparse. From the perspective of distribution, as samples move towards sparse regions, they are stepping out of the main mass of the training distribution and are considered as out-of-distribution (OoD) samples. This aligns with the general expectation that out-of-distribution data points should have high uncertainties and thus, low confidence estimates.

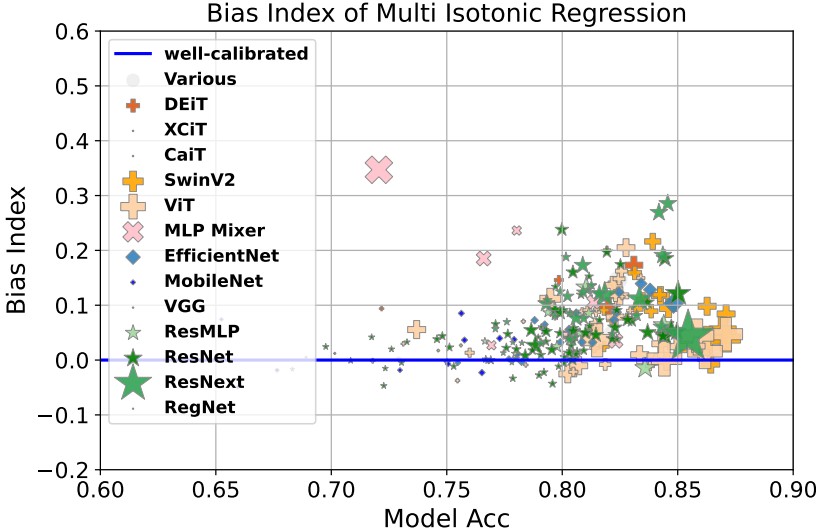

Figure 7: Proximity bias analysis of the model confidence calibrated using **multi isotontic regression** on 504 public models. Each marker represents a model, where marker sizes indicate model parameter numbers and different colors/shapes represent different architectures. The bias index is computed using Equation (3) (0 indicates no proximity bias).

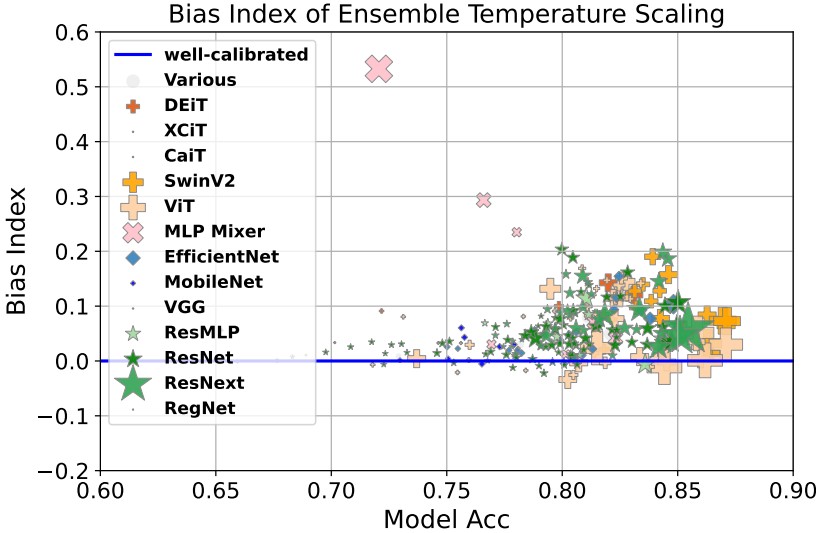

Figure 8: Proximity bias analysis of the model confidence calibrated using **ensemble temperature scaling** on 504 public models. Each marker represents a model, where marker sizes indicate model parameter numbers and different colors/shapes represent different architectures. The bias index is computed using Equation (3) (0 indicates no proximity bias).

However, Figure 10 also shows that the slopes of the accuracy and confidence change are not the same, resulting in a larger miscalibration gap between low proximity and high proximity samples. Furthermore, commonly used calibration techniques such as temperature scaling and histogram binning seems not alleviate this issue. This motivated us to study how proximity information relates to calibration.

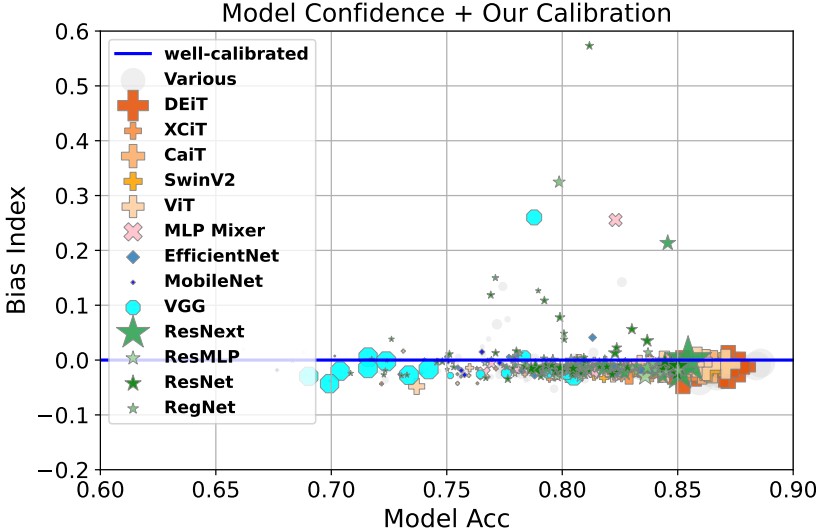

Figure 9: Proximity bias analysis of the model confidence calibrated using our proposed PROCAL on 504 public models. Each marker represents a model, where marker sizes indicate model parameter numbers and different colors/shapes represent different architectures. The bias index is computed using Equation (3) (0 indicates no proximity bias).

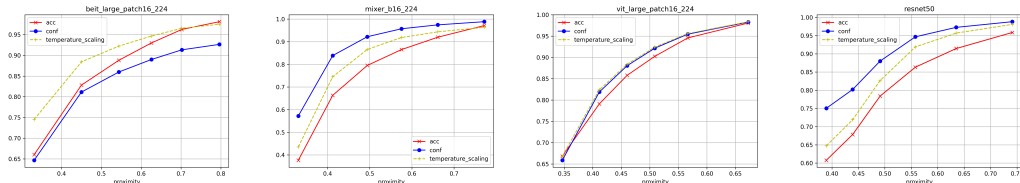

Figure 10: Relations between proximity and its corresponding accuracy, confidence and calibrated confidences. The result shows that confidence and accuracy drop as the sample's proximity decreases.

## E   Additional Experimental Results

### E.1   Inference Efficiency

**Runtime Efficiency**    To verify the time efficiency of our method, we compare the inference time with baseline methods. The result is reported in Table 3. Compared to the confidence baseline, our method, Bin-Mean-Shift, exhibits a slight increase of 1.17% in runtime, while Density-Ratio introduces a modest overhead of 12.3%. These results demonstrate that our method incurs minimal computational overhead while achieving comparable runtime efficiency to the other baseline methods. In addition, it is worth noting that the cost of computing proximity has been reduced due to the recent advancement in neighborhood search algorithms. In our implementation, we employ *indexFlatL2* from the Meta open-sourced GPU-accelerated Faiss library [18] to calculate each sample's nearest neighbor. This algorithm enables us to reduce the time for nearest neighbor search to approximately 0.04 ms per sample (shown in Table 3). The computation overhead beyond the neighbor search is actually quite similar to isotonic regression (IR) and histogram binning (HB), which leads to the total time being roughly twice that of isotonic regression ($0.04 + 0.05 \approx 0.1s$).

**Memory Efficiency**    Similar to K-nearest-neighbor-based methods [17, 45], our method requires maintaining a held-out neighbor set for proximity computation during inference. To achieve memory efficiency, we employ these three techniques: 1) Reduce the size of the held-out neighbor set since it is unnecessary to utilize the entire raw neighbor-set, such as the entire training dataset, particularly when dealing with large-scale data in practical scenarios. For instance, our experimental results on ImageNet [7] are obtained using a neighbor-set comprising 25,000 randomly sampled images

Table 3: Average inference time(ms) per image on ImageNet on 10 runs using a ViT/B-16@224px model on a single Nvidia GTX 2080 Ti. ∗ denotes our method (BIN*: Bin Mean-Shift; DEN*: Density-Ratio Calibration). NS indicates the computational time consumed by the nearest neighbor search algorithm.

| | NS | Conf | TS | ETS | PTS | PTSK | HB | IR | MIR | BIN* | DEN* |
|---|---|---|---|---|---|---|---|---|---|---|---|
| Time(ms) | 0.04 | 5.60 | 5.60 | 5.64 | 5.61 | 5.66 | 5.63 | 5.65 | 5.84 | 5.70 | 6.29 |

from the validation set. 2) Leverage pre-computed feature embeddings instead of full images, such as ResNext101 embeddings ($d = 1024$), which consumed a mere 229MB in our experiments. 3) Leverage memory-efficient neighbor search algorithms to further enhance memory efficiency [34, 19].

### E.2   Effectiveness on Datasets with Balanced Class Distribution.

First, we present high-resolution figures of Figure 3 illustrating the results of 504 models from timm [43] on ImageNet. In Figure 11, our method (indicated by red color markers) consistently demonstrates the lowest calibration error across all four evaluation metrics, maintaining a consistently superior performance.

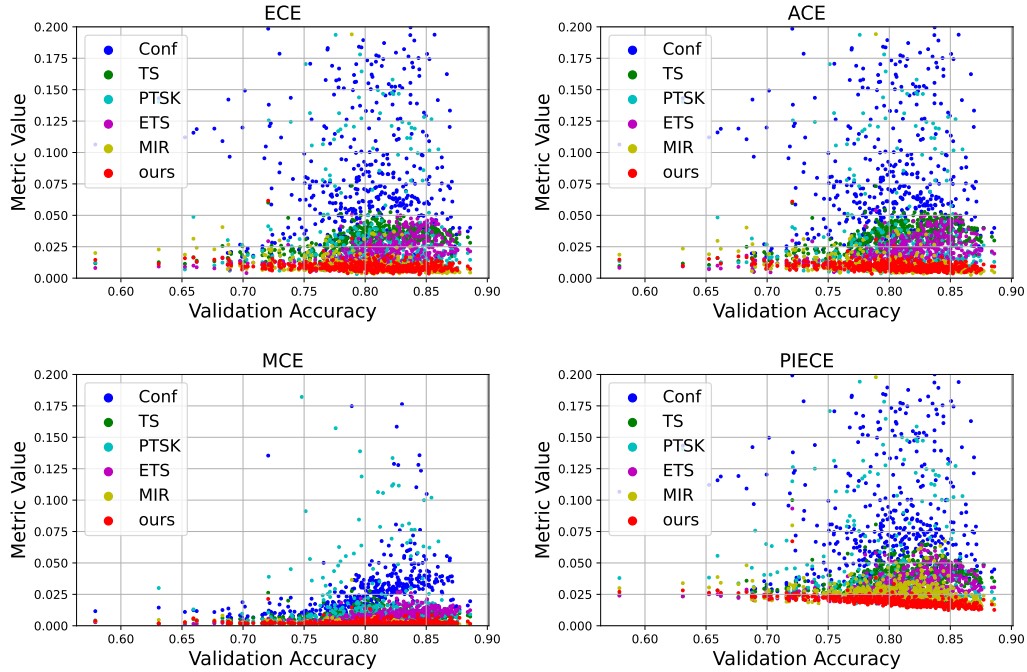

Figure 11: Calibration errors on ImageNet across 504 `timm` models. Each dot represents the calibration results of applying a calibration method to the model confidence. Marker colors indicate different calibration algorithms used. Among all calibration algorithms, our method consistently appears at the bottom of the plot.

Second, we select four widely used ImageNet pre-trained models from the pool of 504 models depicted in Figure 11. We compare the performance of our proposed approach with baseline calibration algorithms, namely `BeiT`, `MLP Mixer`, `ResNet50` and `ViT`. The detailed results are presented in Table 4, showcasing the effectiveness of our methods in mitigating proximity bias and enhancing calibration performance compared to existing calibration techniques. Additionally, even when applied to the most successful base calibration algorithm, our method achieves a notable reduction in calibration errors. This consistent improvement is particularly remarkable, especially in scenarios where the original calibration methods fail to enhance or even worsen performance.

Table 4: Comparison of calibration errors in $10^{-2}$ between existing calibration methods ('base') and our proximity-informed framework ('ours'), on `ImageNet` dataset. ($*$ means $p = 0.01$)

| Model | Method | ECE ↓ | | ACE ↓ | | MCE ↓ | | PIECE ↓ | |
|---|---|---|---|---|---|---|---|---|---|
| | | base | +ours | base | +ours | base | +ours | base | +ours |
| BeiT | Conf | 3.6137 | **0.8573*** | 3.5464 | **0.7205*** | 1.5801 | **0.2866*** | 4.2348 | **1.5379*** |
| | ETS | 2.1318 | **0.9930*** | 2.1862 | **0.9023*** | 1.2155 | **0.4333*** | 2.9592 | **1.5872*** |
| | HB | **4.8765*** | 5.5631 | 6.1728 | **5.9747** | **1.8383*** | 4.1239 | 7.2174 | **6.2886*** |
| | MIR | **0.4509** | 0.536 | **0.5376** | 0.5455 | **0.1065** | 0.1395 | 1.8039 | **1.2645*** |
| | PTS | 1.2685 | **0.9787** | 1.2744 | **0.8106** | 0.4858 | **0.4240** | 1.9782 | **1.6890** |
| | PTSK | 1.8861 | **1.0288** | 1.9150 | **0.8582*** | 0.7934 | **0.5022** | 2.7093 | **1.6168*** |
| | TS | 2.9894 | **1.3277*** | 3.1132 | **1.2493*** | 0.7388 | **0.7046** | 3.5366 | **1.9264*** |
| Mixer | Conf | 10.9366 | **2.8498*** | 10.9337 | **2.7162*** | 5.1519 | **1.0944*** | 11.0164 | **3.6485*** |
| | ETS | 1.9381 | **1.3586*** | 2.1859 | **1.2210*** | 0.3204 | **0.2986** | 4.1034 | **2.2010*** |
| | HB | 9.1774 | **6.7207*** | 9.7965 | **7.4825*** | **3.5964*** | 4.3997 | 12.9240 | **7.8672*** |
| | MIR | 1.1128 | **0.9272** | 1.2190 | **0.9360*** | 0.2628 | **0.1430*** | 3.3912 | **2.2112*** |
| | PTS | 5.7741 | **2.2208** | 5.8027 | **2.1298** | 2.8498 | **0.6940** | 9.3215 | **3.0623*** |
| | PTSK | 6.6610 | **1.9466*** | 6.6173 | **1.7905*** | 3.1011 | **0.6131*** | 8.3148 | **2.7503*** |
| | TS | 5.1937 | **1.6499*** | 5.0234 | **1.4455*** | 2.0189 | **0.4255*** | 5.8958 | **2.4809*** |
| ResNet50 | Conf | 8.7246 | **2.7752*** | 8.6852 | **2.6344*** | 4.6122 | **1.3005*** | 8.9113 | **3.4224*** |
| | ETS | 2.7620 | **1.6548*** | 3.6581 | **1.6627*** | 0.6624 | **0.5676** | 3.5750 | **2.4579*** |
| | HB | 7.6311 | **6.1812*** | 9.3289 | **7.5380*** | **2.6377*** | 4.3484 | 10.1849 | **7.7372*** |
| | MIR | 1.0281 | **0.9533** | 0.9643 | **0.8436** | **0.2062** | 0.2095 | 1.9751 | **1.8776*** |
| | PTS | 2.2196 | **1.0138*** | 2.2098 | **1.0278*** | 0.6319 | **0.2529** | 4.0331 | **2.0445*** |
| | PTSK | 4.4100 | **1.6015*** | 4.3761 | **1.5015*** | 1.8375 | **0.4980** | 5.4413 | **2.5003*** |
| | TS | 5.1181 | **1.7964*** | 5.0864 | **1.7300*** | 2.5503 | **0.6881*** | 5.4640 | **2.5721*** |
| ViT | Conf | 1.1815 | **0.9016*** | 1.1839 | **0.7554*** | 0.3489 | **0.2543** | 1.9984 | **1.7540*** |
| | ETS | 1.1080 | **0.9074** | 1.1791 | **0.7732*** | 0.2745 | **0.2684** | 1.9273 | **1.7533** |
| | HB | **4.6920*** | 6.8776 | **7.1771** | 7.5824 | **2.4332*** | 4.8925 | **7.3239*** | 7.7552 |
| | MIR | 0.8934 | **0.8638** | **0.8180** | 0.8537 | **0.1893** | 0.2182 | 1.7695 | **1.6818** |
| | PTS | 1.0609 | **0.7940** | 1.0257 | **0.7124** | 0.3937 | **0.2600** | 2.0929 | **1.7510*** |
| | PTSK | 1.6493 | **0.8966** | 1.5874 | **0.8139** | 0.6035 | **0.2957** | 2.5798 | **1.8211** |
| | TS | 1.4905 | **0.9047*** | 1.4465 | **0.7880*** | 0.5069 | **0.2806*** | 2.1462 | **1.7562*** |

Third, we present the outcomes of our approach on the Natural Language Understanding task, specifically on the MultiNLI Match dataset, as displayed in Table 5. The results demonstrate that our method can improve confidence calibration performance in balanced datasets and achieve comparable performance in addressing proximity bias.

Table 5: Results of MultiNLI Match dataset on RoBERTa-base Model that is fine-tuned on MultiNLI Match. 'Base' refers to existing calibration methods and 'Ours' refers to our method applied to existing calibration methods. Calibration error is given by $\times 10^{-2}$.

| Method | ECE ↓ | | ACE ↓ | | MCE ↓ | | PIECE ↓ | |
|---|---|---|---|---|---|---|---|---|
| | base | +ours | base | +ours | base | +ours | base | +ours |
| Conf | 2.39 | **1.71** | 2.68 | **2.03** | 0.54 | **0.36** | 4.07 | **3.70** |
| TS | 1.68 | **1.53** | 1.96 | **1.86** | 0.49 | **0.40** | 4.19 | **3.45** |
| ETS | 1.88 | **1.57** | 2.08 | **1.76** | 0.70 | **0.44** | 3.97 | **3.61** |
| PTS | 9.28 | **3.48** | 9.25 | **3.49** | 5.86 | **1.13** | 9.44 | **5.29** |
| PTSK | 11.94 | **6.10** | 11.91 | **6.15** | 10.24 | **4.03** | 12.23 | **6.80** |
| HB | 2.07 | **1.80** | 2.10 | **2.10** | 1.03 | **0.67** | 4.63 | **3.67** |
| IR | 1.30 | **1.23** | 1.71 | **1.54** | **0.30** | 0.45 | 3.70 | **3.66** |
| MIR | **1.02** | 1.04 | 1.22 | **1.22** | 0.34 | **0.32** | 3.73 | **3.35** |

## E.3 Effectiveness on Datasets with the Long-tail Class Distribution.

To evaluate our method in large-scale long-tail datasets, we conduct experiments on long-tail datasets `ImageNet-LT`, and `iNaturalist 2021`. Table 6 shows our method's performance on ImageNet-LT, showing that our algorithm improves upon the original calibration algorithms in most cases under all four evaluation metrics, particularly on ECE, ACE, and PIECE. This suggests that our algorithm

Table 6: Performance of our proposed framework against base methods on ImageNet-LT using the pretrained ResNet50 model with classifier re-training techniques [20]. The symbol $*$ denotes that the method is significantly better than the other one with a confidence level of at least 90%.

| Method | ECE ↓ base | ECE ↓ +ours | MCE ↓ base | MCE ↓ +ours | ACE ↓ base | ACE ↓ +ours | PIECE ↓ base | PIECE ↓ +ours |
|---|---|---|---|---|---|---|---|---|
| conf | 7.4933 | **1.8724*** | 0.8672 | **0.3380*** | 7.4610 | **2.0168*** | 7.9229 | **3.9776*** |
| TS | 2.1965 | **1.8541** | **0.3100** | 0.3956 | 2.0580 | **1.7472** | **3.8407** | 3.8478 |
| ETS | 2.2704 | **1.8642** | **0.3264** | 0.3926 | 2.0648 | **1.7622** | 4.0024 | **3.8520** |
| PTS | 5.0023 | **1.7300*** | 0.6512 | **0.3160** | 4.9927 | **1.7401*** | 7.6502 | **3.9018*** |
| PTSK | 9.1862 | **1.8558** | 1.7478 | **0.3475** | 9.2080 | **1.9875** | 11.4242 | **3.9175*** |
| HB | 13.4644 | **12.9257*** | 7.5672 | **7.5670** | 13.5931 | **13.4697** | 16.2915 | **15.3259*** |
| IR | **8.0541** | 8.5778 | 1.0642 | **1.0508** | **7.9947** | 8.5546 | **8.8631** | 9.3678 |
| MIR | 1.6207 | **1.5063** | **0.3205** | 0.3217 | 1.6562 | **1.5310** | 3.7481 | **3.6285** |

can effectively mitigate the bias towards low proximity samples (i.e. tail classes), highlighting its practicality in real-world scenarios where data is often imbalanced.

### E.4 Effectiveness on Datasets with Distribution Shift

To further evaluate the effectiveness of our proposed method in handling distribution shifts, we conduct experiments on the ImageNet Corruption dataset [14]. The base models are trained on the ImageNet dataset without any distribution shifts and are subsequently tested on the corrupted dataset. The results, shown in Figure 12, indicate that our method is able to consistently improve the performance of the original model even in the presence of distribution shifts. This is particularly notable in the case of data with distribution shifts that tend to fall in the low proximity region, where current algorithms tend to be poorly calibrated, but our method effectively addresses this problem.

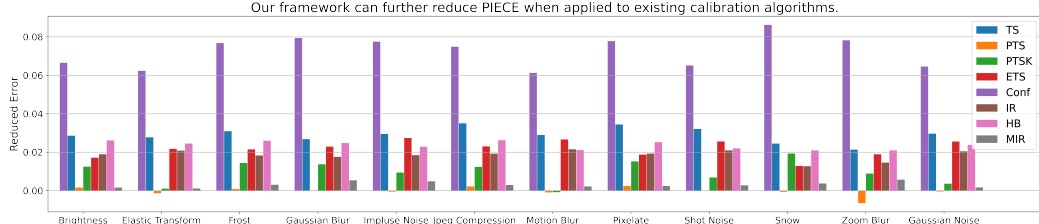

Figure 12: The calibration error reduction in PIECE aachieved by integrating our method with existing calibration algorithms. Different colors indicate different base calibration algorithms. Each color represents a different base calibration algorithm. The bar indicates the difference in calibration error between the base algorithm and the one enhanced by our approach.

To explore the issue of domain shift, where test data has a shifted distribution not seen in training, we conduct experiments using ImageNet (training set) and ImageNet-Sketch (test set). The datasets are chosen because all ImageNet images are real-world photos, while all ImageNet-Sketch images are sketches, collected using Google Search with class label keywords and "sketch", similar to the case of skin color example provided in the introduction. We employ a Vision Transformer backbone from TIMM, trained on the ImageNet. Then we train our ProCal using the validation set from ImageNet and tested it on the 50,000 images from ImageNet-Sketch. The result shown in Table 7 shows that ProCal effectively improves upon existing algorithm in many cases. While we observe a slight increase in ECE, ACE, and MCE when ProCal is paired with with PTS and PTSK, this is probably attributed to the original methods suffering from the *cancellation effect*, where positive and negative calibration errors within the same confidence bin cancel out each other (see section 6.1). Under the PIECE metric that captures the cancellation effect, our method consistently outperforms all methods by large margins and effectively mitigates their proximity bias.

Table 7: Calibration performance of ViT-Large (vit_large_patch14_clip_336) from timm [43] on ImageNet-Sketch dataset. Calibrators are trained using ImageNet validation set and tested on ImageNet-Sketch. 'base' refers to the methods, '+ours' shows the performance after integrating our method. Note that 'Conf+Ours' shows the result of our method applied directly to model confidence. Calibration error is given by $\times 10^{-2}$.

| Method | ECE ↓ base | ECE ↓ +ours | ACE ↓ base | ACE ↓ +ours | MCE ↓ base | MCE ↓ +ours | PIECE ↓ base | PIECE ↓ +ours |
|---|---|---|---|---|---|---|---|---|
| Conf | 3.91 | **1.96** | 3.92 | **1.92** | 0.95 | **0.35** | 6.33 | **2.92** |
| TS | 7.60 | **2.38** | 7.57 | **2.38** | 1.42 | **0.57** | 7.95 | **3.27** |
| ETS | 3.16 | **2.07** | 3.22 | **2.03** | **0.37** | 0.39 | 4.92 | **2.97** |
| PTS | 1.62 | **1.60** | **1.65** | 1.66 | 0.34 | **0.25** | 3.89 | **2.74** |
| PTSK | 3.14 | **1.15** | 3.16 | **1.11** | 0.96 | **0.13** | 5.71 | **2.58** |
| MIR | **0.28** | 1.22 | **0.23** | 1.19 | **0.12** | 0.22 | 4.24 | **2.73** |

# F    Ablation Study

## F.1    Comparison between Density-Ratio and Bin-Mean-shift.

Our comparison of Density-Ratio and Bin-Mean-shift reveals that Density-Ratio performs better in scaling-based methods while Bin-Mean-Shift demonstrates general robustness and adaptability in both continuous and discrete settings. This can be attributed to the fact that Density-Ratio can be thought of as an infinite binning-based method, enjoying good expressiveness, but it relies on density estimation which may not be as accurate when dealing with discrete outputs. In contrast, Bin-Mean-shift does not make any assumptions about the output. Therefore, it is important to note that both techniques have their own strengths and weaknesses, and the choice of which one to use should be based on the specific task at hand.

## F.2    Hyperparameter Sensitivity

In this section, we evaluate the sensitivity of our method to various hyperparameter choices. Specifically, we examine the impact of the choice of distance metric and the number of neighbors in the local neighborhood. To evaluate the performance of our method under different hyperparameter settings, we use ResNet50 [12] as our base model and compare its calibration performance of the integration of our method and existing popular claibration algorithms under different hyperparameters. This study aims to provide a guideline for selecting appropriate hyperparameters when using our method.

**Effect of Neighbor Size $K$**    In this study, we examine the influence of the number of neighbors ($K$) on the performance of our proposed method. We assess the performance by varying the number of neighbors from $K = 1$ to $K = 1000$ and comparing the results. Figure 13 illustrates the impact of neighbor size $K$ on performance. The results reveal a V-shaped relationship between the number of neighbors and performance. Initially, an increase in the number of neighbors from 1 yields increasing performance improvement. However, when the number of neighbors exceeds a certain threshold (e.g. $K > 50$), performance begins to deteriorate as the neighborhood becomes more global rather than local, eventually reaching a saturation point. Notably, a small neighborhood size of $K = 10$ is sufficient to capture the local neighborhood, and further increasing the neighborhood size does not yield additional benefits. This finding aligns with previous works [17, 45] that demonstrate the insensitivity of proximity to the choice of $K$. Based on these findings, we recommend employing a moderate range of neighbors, specifically between 10 and 50, to achieve optimal performance.

**Effect of Distance Measure**    In this study, we explore the impact of different distance measures on the performance of our method. We compare the performance under four distance measures: L2 (Euclidean distance), cosine similarity, IVFFlat, and IVFPQ. L2 distance and cosine similarity are widely used measure in machine learning. IVFFlat and IVFPQ are approximate nearest-neighbor search methods implemented using the `faiss` library [18]. IVFFlat is memory-efficient and suitable for high-dimensional datasets, while IVFPQ is optimized for datasets with a large number of points and high-dimensional features. The results are depicted in Figure 14. Overall, we observe minimal performance differences across the various distance measures. Specifically, the use of cosine similarity

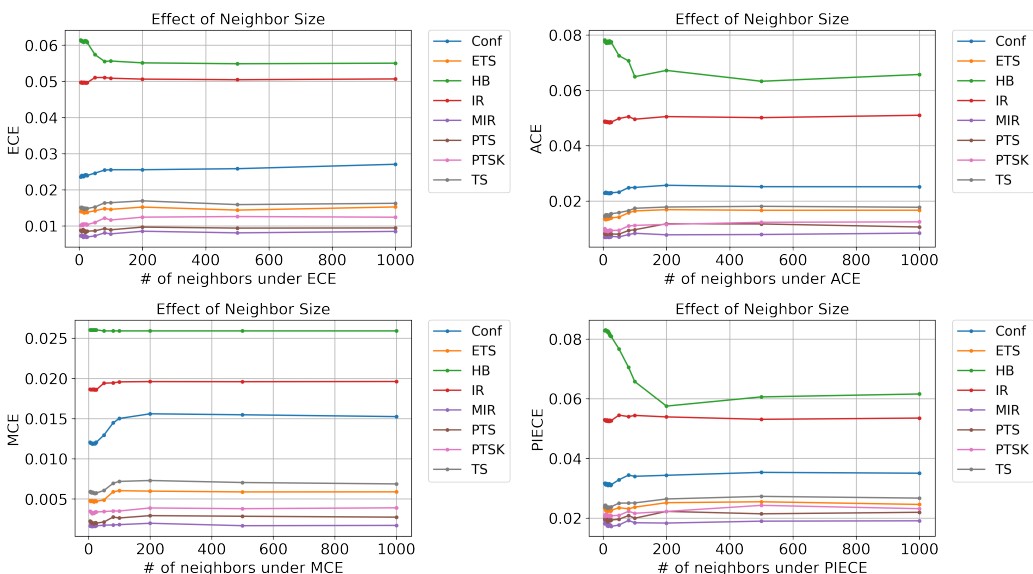

Figure 13: Hyperparameter sensitivity of the number of neighbors used in computing proximity. The results reveal a V-shaped relationship between the number of neighbors and performance. Initially, an increase in the number of neighbors from 1 yields increasing performance improvement. However, when the number of neighbors exceeds a certain threshold (e.g. $K > 50$), performance begins to deteriorate.

and L2 distance yields comparable performance across the four calibration metrics. Additionally, employing IVFFlat results in slightly smaller calibration errors. However, the performance disparity is not significant. Different distance measures, including the approximation methods, offer similar performance improvements. This is because the density function for proximity values is smooth, and small measurement noise does not significantly impact the final density estimation. Considering efficiency, we recommend utilizing IVFFlat due to its favorable efficiency characteristics. However, it is important to note that the choice of the best distance measure depends on the specific problem and dataset, as each measure may exhibit varying performance in different scenarios.

# G Pseudo-codes

We present the procedural steps of our approach in the form of a pseudocode. Algorithm 1 encompasses the general inference phase:

---

**Algorithm 1** Inference procedure.

---

**Require:** Test sample $\mathbf{x} \in \mathbb{R}^n$, held-out embeddings $E \in \mathbb{R}^{N \times d}$, classifier $f$ (calibrated or uncalibrated), number of nearest neighbors $K$, nearest neighbor search algorithm $S$ using `Faiss` library, PROCAL calibrator $C$

1: **procedure** INFERENCE($\mathbf{x}$)
2:      $\mathbf{e}_x, \hat{p}, \hat{y} \leftarrow f(\mathbf{x})$                    ▷ get feature embedding, prediction and confidence
3:      $\mathbf{d} \leftarrow S.search(\mathbf{e}_x, K)$
4:      $d_x \leftarrow exp\{-mean(\mathbf{d})\}$        ▷ proximity as the average distance to $K$ nearest neighbors
5:      **return** $C(\hat{p}, d_x)$                               ▷ return calibrated confidence
6: **end procedure**

---

Algorithm 2 encapsulates the Density-Ratio Calibration algorithm.

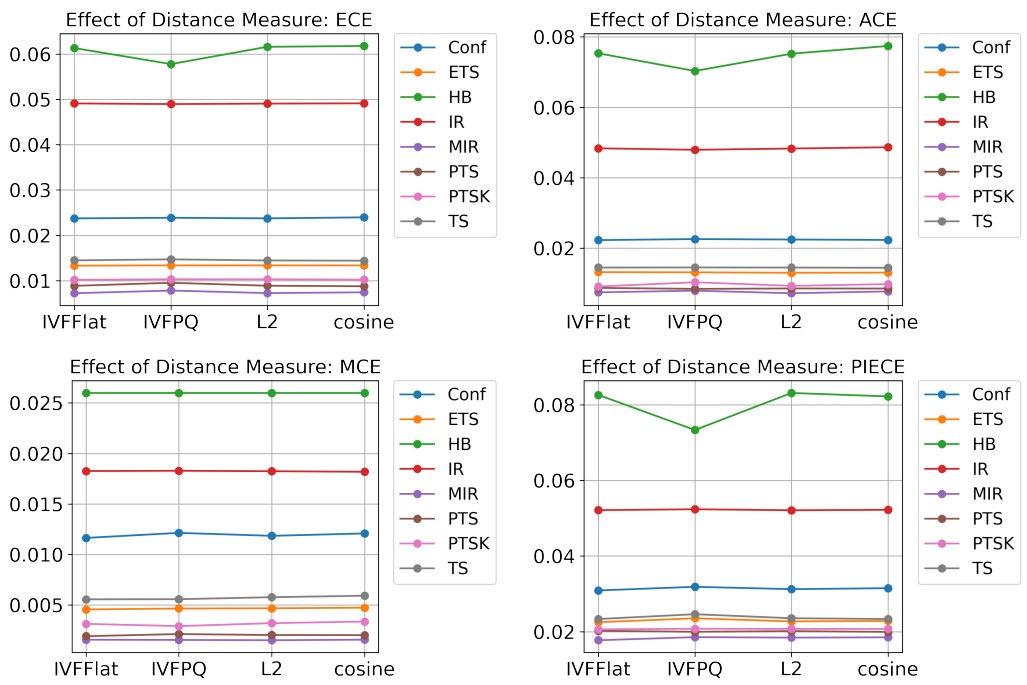

Figure 14: Hyperparameter sensitivity of several proximity measure under four evaluation metrics.

---

**Algorithm 2** Density-Ratio Calibration.

---

**Require:** Pre-trained model $M$, validation set with pre-computed proximity $\mathcal{D}_{val} = \{\mathcal{X}, \mathcal{Y}, \mathcal{D}\}$, test set with pre-computed proximity $\mathcal{D}_{test} = \{\mathcal{X}_{test}, \mathcal{Y}_{test}, \mathcal{D}_{test}\}$,

1: **procedure** DENSITYRATIOCALIB(**x**)
2:     $\mathcal{D}_{val}^{+} = \emptyset, \mathcal{D}_{val}^{-} = \emptyset$
3:     **for** $i = 1, \ldots, |\mathcal{X}|$ **do**
4:         $\hat{y}_i, p_i \leftarrow M(x_i), x_i \in \mathcal{X}$                        ▷ get predicted class label and confidence
5:         **if** $\hat{y}_i = y$ **then**                            ▷ split $\mathcal{D}_{val}$ based on prediction correctness
6:             $D_{val}^{+} \leftarrow \mathcal{D}_{val}^{+} \cup \{<p_i, d_i>\}$
7:         **else**
8:             $\mathcal{D}_{val}^{-} \leftarrow \mathcal{D}_{val}^{-} \cup \{<p_i, d_i>\}$
9:         **end if**
10:     **end for**
11:     KDE$^{+} \leftarrow KDE(\mathcal{D}_{val}^{+})$              ▷ 2-dimension KDE given confidence and proximity
12:     KDE$^{-} \leftarrow KDE(\mathcal{D}_{val}^{-})$
13:     $\gamma \leftarrow \frac{|\mathcal{D}_{val}^{-}|}{|\mathcal{D}_{val}^{+}|}$
14:     **for** $j = 1, \ldots, |\mathcal{X}_{test}|$ **do**
15:         $\hat{y}_j, p_j \leftarrow M(x_j), x_j \in \mathcal{X}_{test}$
16:         $s_j = \frac{\text{KDE}^{+}(D_j, p_j)}{\text{KDE}^{+}(D_j, p_j) + \gamma \times \text{KDE}^{-}(D_j, p_j)}$     ▷ compute re-calibrated score for test sample
17:     **end for**
18:     **return** $s_j, j = 1, \ldots, |\mathcal{X}_{test}|$
19: **end procedure**

---

