# OpenReview forum: "Proximity-Informed Calibration for Deep Neural Networks"
_NeurIPS.cc/2023/Conference — NeurIPS 2023 spotlight_

### Official Review · Reviewer_4HP9 · 2023-06-29

**Soundness:** 3 good
**Presentation:** 4 excellent
**Contribution:** 3 good
**Rating:** 6
**Confidence:** 3

**Summary:**

This paper quantifies and proposes a mitigation for a phenomenon in DNN training, where more 'unusual' examples (here defined as having a higher average distance to its K=10 nearest neighbors) are generally more miscalibrated across a range of models and tasks. The authors propose a new proximity-aware calibration metric, PIECE, and demonstrate that it can capture calibration issues that are not necessarily captured by the standard ECE metric (and, in fact, always P>= ECE). Additionally, the authors propose a mitigation, ProCal. This mitigation is presented in two variations, one for continuous and one for discrete confidence. These work by adjusting the uncalibrated probability score based on the model's miscalibration on examples with that average distance.

**Strengths:**

The paper is very clearly written, with a logical flow and good explanations of all steps taken. I particularly enjoyed the motivation via experiments on existing models leading to theory-driven proposals leading to further experimental verification. I also appreciated the through ablation study and the extra attention given to OOD examples.

In addition, the original miscalibration problem that the paper brings up seems to me to be important and relevant, and is well-motivated. As I am not an expert in miscalibration mitigation techniques I cannot fully comment on the novelty aspect of the work, though I have not seen this 'atypicality' concern brought up explicitly before; it also dovetails neatly with other research in atypical examples.

The proposed metric and mitigations, while very simple, are logical and effective, which I hope will lead to their adoption.



**Weaknesses:**

I felt that the 'atypical' (high $D(X)$) examples could have been better characterized. In particular, the authors relate these examples to underrepresented categories in datasets (eg, Black people in health datasets), but this is not necessarily so. It does not seem incredible to have a scenario where a data is divided into two clusters, where the smallest cluster is nevertheless very tight, and so has a smaller average $D(X)$.

It is not really clear what the ProCAL method refers to. It seems like it mostly splits into two methods each of which have their own name. Maybe it would be better to call them ProCAL-C and ProCAL-D  (for continuous and discrete)?

The analysis of ProCAL effectiveness seems a little sparse in claiming "our method consistently improves the calibration"... My interpretation of the data would be that ProCAL is very helpful in conjunction with raw confidence as well as some of the other methods with higher PIECE scores (TS variants), but not as effective for methods such as IR and MIR, where it seems to hurt as much as help. I don't think that this disqualifies the paper, but a more thorough explanation and analysis of this would be appropriate.

(minor) in section 3.1, what is the sensitivity of two points having the 'same' confidence?
(minor) It is not clear why the definition of D(X) contains an exponent, rather than the simple average of the ten smallest distances.


**Questions:**

Please see the weakness section above. In addition:

* For cases where the test distribution doesn't match the train distribution, how much are we losing by using the test data to estimate values of D (versus measuring the distance of a test sample point to the closest 10 training points)? It would seem that the training distribution is actually the important one for estimating atypicality.
* How does the Bias Index changed if only $B_H$ is used for sampling points and $B_L$ is used for matching, and vice versa? What if the intermediate groups are used?
* Is it possible to characterize the $B_H$ and $B_L$ groups? Is there a large difference of average confidence between the two, and how well do their confidences intersect?
* Why is ProCAL less effective in conjunction with IR and MIR?


**Limitations:**

The limitations are adequately addressed.

---

> ### Author Rebuttal · Authors · 2023-08-09
>
> We appreciate reviewer 4HP9's constructive feedback. We are glad the reviewer enjoyed the logic flow of this paper, finds the problem important and the solution simple, logical and effective.
>
> **Q1: I felt that the 'atypical' (low proximity) examples could have been better characterized. In particular, the authors relate these examples to underrepresented categories in datasets (eg, Black people in health datasets), but this is not necessarily so. It does not seem incredible to have a scenario where the data is divided into two clusters, where the smallest cluster is nevertheless very tight, and so has a smaller average.**
>
> This is indeed a thoughtful question. First, we want to clarify that 'low proximity' in our definition refers to 'low density', but the context of 'underrepresented' here links more to 'low probability', which are correlated but not equivalent.
>
> Second, we agree with the reviewer that our existing definition of proximity does not include samples with low probability but high density.
> It is still an open question whether these samples underrepresented but with high proximity suffer from proximity bias (i.e., are low probability samples inherently more overconfident than high probability samples) and whether they need mitigation. We leave the investigation of these cases and a better characterization to the future work.
>
> **Q2: It is not really clear what the ProCAL method refers to. It seems like it mostly splits into two methods each of which have their own name. Maybe it would be better to call them ProCAL-C and ProCAL-D (for continuous and discrete)?**
>
> Thank you for your suggestion. The intention behind the term 'ProCAL' is to represent the general strategy and a series of methods for mitigating proximity bias. Since we've tailored two specific methods for different output types, we've referred to these as Density-Ratio Calibration and Bin-Mean-Shift. We will clarify this in the paper.
>
> **Q3: why the definition of D(X) contains an exponent, rather than the simple average of the ten smallest distances.**
>
> This is because the current definition normalizes the distance measure from a range of $[0, \inf]$ to $(0,1]$, making the approach more robust to the effects of distance scaling since the absolute distance in Euclidean distance can cause instability and difficulty in modeling.
>
> **Q4: For cases where the test distribution doesn't match the train distribution, how much are we losing by using the test data to estimate values of D**.
>
> First, we want to clarify that we actually use a validation set rather than a test set for estimation atypicality (i.e., proximity), which we assume is iid drawn from the training distribution.  Besides, in most cases, including settings with distribution shift, we do not presume to have knowledge of the distribution-shifted data, and therefore we have no access to the test data.
>
> However, in case we can access the test dataset to estimate the proximity, how much we are losing is largely dependent on how far the test distribution deviates from the training distribution.
>
> **Q5: Why is ProCAL less effective in conjunction with IR and MIR?** My interpretation of the data would be that ProCAL is very helpful in conjunction with raw confidence as well as some of the other methods with higher PIECE scores (TS variants), but not as effective for methods such as IR and MIR, where it seems to hurt as much as help. I don't think that this disqualifies the paper, but a more thorough explanation and analysis of this would be appropriate.
>
> There are two cases related to the question:
> 1. **ProCal improves PIECE but negatively impacts ACE/ECE**:  we argue that the "cancellation effect" (see lines 170-181) is responsible for this. This effect emerges when positive calibration errors (i.e., acc - conf > 0) from high proximity samples and negative errors from low proximity samples offset each other within the same confidence group, causing ACE/ECE to potentially underestimate the calibration error. Binning-based methods are particularly prone to this because it shifts all samples in a bin to the same confidence, increasing the chances of the cancellation effect. Although ProCAL, which corrects each proximity bin independently, might increase ECE/ACE by disrupting this cancellation effect, a rise in ECE/ACE doesn't necessarily reflect overall calibration degradation.
> 2. **ProCal worsens PIECE**: note that IR/MIR are binning-based and hence Bin-Mean-Shift is used. We speculate that the deterioration in Bin-Mean-Shift stems from the inaccuracies of each bin's estimates. When bins have few samples, their accuracy estimates can vary significantly and hence unreliable, leading to this observed performance decline.
>
> ---
>
> For questions pertaining to the computation of the bias index, we direct the reviewer to our **global response** above.

---

> > ### Comment · Reviewer_4HP9 · 2023-08-11
> > **Thank you for your response**
> >
> > I thank the reviewers for the responses to my and other reviewers' questions and suggestions. It seems that on the whole the final paper will be quite similar to the version we reviewed with some additional clarifications.
> >
> > I do think the question of domain shift could be better explored (or explained). For a concrete real-world example, to use the skin color/cancer detection example. If the samples in the training data are collected from majority light-skinned individuals, the ProCAL method would likely help adjust the confidences for the dark-skinned minority. However, if the training data are collected from exclusively light-skinned individuals, can the ProCAL method still be effective?
> >
> > Overall, I keep my score, and thank the authors for an interesting paper.

---

> > > ### Author Response · Authors · 2023-08-15
> > > **Experiments show that ProCal effectively improves upon existing algorithms in the domain shift setting**
> > >
> > > Thank you for the prompt reply and the insightful question!
> > >
> > > To explore the issue of domain shift, where test data has a shifted distribution not seen in training, we conduct experiments using **ImageNet (training set)** and **ImageNet-Sketch (test set)**. The datasets are chosen because all ImageNet images are real-world photos, while all ImageNet-Sketch images are sketches, collected using Google Search with class label keywords and "sketch", similar to the case of skin color example.
> > >
> > > **Experiment Setup**: We employ a ResNet50 backbone from `timm`, trained on the ImageNet. Then we train our ProCal using the validation set from `ImageNet` and tested it on the 50,000 images from `ImageNet-Sketch`. The term `base` represents the corresponding method's outputs (e.g. Conf, TS, ETS), while "+ours" indicates the output confidence scores corrected using our approach.
> > >
> > > | Method | ECE | ACE | MCE | PIECE |
> > > | :--- | :--- | :--- | :--- | :--- |
> > > |  | base / +ours | base / +ours | base / +ours | base / +ours |
> > > | Conf| 0.0871 / **0.0176** | 0.0869 / **0.0174** | 0.0365 / **0.0062** | 0.0885 / **0.0255** |
> > > | TS  | 0.0501 / **0.0180** | 0.0501 / **0.0170** | 0.0153 / **0.0025** | 0.0524 / **0.0223** |
> > > | ETS | 0.0457 / **0.0192** | 0.0479 / **0.0192** | 0.0067 / **0.0024** | 0.0494 / **0.0221** |
> > > | PTS | **0.0126** / 0.0129 | **0.0125** / 0.0132 | **0.0019** / 0.0023 | 0.0208 / **0.0188** |
> > > | PTSK| **0.0094** / 0.0114 | **0.0096** / 0.0126 | **0.0011** / 0.0019 | 0.0275 / **0.0191** |
> > > | MIR | 0.0194 / **0.0154** | 0.0193 / **0.0155** | 0.0032 / **0.0030** | 0.0243 / **0.0240** |
> > >
> > >
> > > **Results**: As demonstrated in the table above, ProCal effectively improves upon existing algorithm in many cases. While we observe a slight increase in ECE/ACE/MCE when ProCal is paired with with PTS and PTSK, this is probably attributed to the original methods suffering from the "cancellation effect”, where positive and negative calibration errors within the same confidence bin cancel out each other (see responses to Q5 above; or lines 170-181). Under the PIECE metric that captures the cancellation effect, our method consistently outperforms all methods by large margins, and effectively mitigates their proximity bias.

---

### Official Review · Reviewer_4gc8 · 2023-07-05

**Soundness:** 3 good
**Presentation:** 3 good
**Contribution:** 4 excellent
**Rating:** 7
**Confidence:** 4

**Summary:**

This paper studies the prevalence of proximity bias in calibration, i.e. the rate of miscalibration on samples that are far away from their nearest neighbors in the data ("low proximity"). The authors empirically show that this type of miscalibration is present across many models, and propose a new post-training calibration procedure for mitigating it. Their approach shows significant empirical improvements over standard post-training calibration approaches.

**Strengths:**

- **Originality:** The empirical investigation and proposed methodology in the paper is quite novel, as I am not aware of prior work that has studied this type of bias in calibration (although others have studied subgroup calibration).
- **Quality:** The claims in the paper are technically sound, and the experiments exploring proximity bias are extensive (more than 500 pretrained models considered, and various calibration baselines).
- **Clarity:** Overall, the paper is well-written and well-organized, with motivating experiments and intuitive definitions. However, I believe the presentation of the calibration algorithm in the paper could be improved (detailed further in weaknesses).
- **Significance:** The idea of low proximity samples introduced and analyzed by the authors seems quite significant, as these samples can correspond to underrepresented populations in the data.

**Weaknesses:**

- **Algorithm Details:** The weakest part of the paper in my view is the lack of detail in Section 5.1. This part of the paper would be significantly improved by including something akin to the pseudo-code algorithm in the appendix. There are several questions that arise when reading this part: what does one do after estimating the posterior probability conditional on prediction and proximity (in the algorithm I can see that this is just the output on the test point)? Do we compute proximity only with respect to the test data? What type of KDE is used (i.e. kernel, bandwidth, etc.)?
- **Implementation Details:** In addition to algorithm details, some parts of the experimental setup could also be made clearer. How are ECE and ACE computed (i.e. binning scheme)? Do you set aside calibration data for the scaling methods (TS, ETS, PTS, PTSK) in addition to the set aside data for ProCal?

**Questions:**

The main questions I have are detailed in weaknesses above.

**Limitations:**

The authors appropriately discuss limitations, but it might be helpful to include some bits from the appendix (particularly regarding efficiency) in the main body.

---

> ### Author Rebuttal · Authors · 2023-08-09
>
> We sincerely thank reviewer 4gc8 for the constructive feedback and we are glad that the reviewer finds the investigation and methodology novel, the claim technically sound, the experiment extensive, and the proximity bias issue significant. Here we answer all the questions and hope they can address the concerns.
>
> ---
>
>
> **W1: The weakest part of the paper in my view is the lack of detail in Section 5.1. This paper would be significantly improved by including something akin to the pseudo-code algorithm in the appendix.**
>
> We appreciate your constructive feedback. We will incorporate the pseudo-code algorithms in Section G into the main paper in our forthcoming revision.
>
>
> **Q1: what does one do after estimating the posterior probability conditional on prediction and proximity (in the algorithm I can see that this is just the output on the test point)?**
>
> Thank you for the question. In fact, we directly treat the estimated posterior probability $\mathbb{P}(\hat{Y}=Y \mid \hat{P}, D)$ as the calibrated confidence. The rationale for this is:
>
> 1) The common interpretation of confidence is the probability of the prediction being correct given the sample X, and our posterior probability serves as an estimate of this $\mathbb{P}(\hat{Y}=Y \mid X)$ by using $\hat{P}, D$ as the proxy of sample X (line 206-210) ;
>
> 2) The commonly-used confidence is $\hat{P}$, while the posterior can be regarded as an updated or modified version of the initial $\hat{P}$.
>
>
> **Q2: Do we compute proximity only with respect to the test data?**
>
> If we understand the question correctly, the reviewers is asking (please let us know if you meant the question in a different way): *given a test sample, if the proximity is computed based on the test set*. To address this, we clarify that the proximity for any test sample is computed using the **validation set**. Here are the details:
>
> 1) During inference, when calibrating a test point, we only need to compute this point’s proximity by finding its $K$ nearest neighbors in the held-out **validation** set and no additional proximity computation is required.
>
> 2) When training the calibration approach (using the validation set for training), we calculate every point's proximity to others within the validation set to estimate their proximity values, which are then used for training the density estimators for Density-Ratio Calibration and the binning parameters in Bin-Mean-Shift. Note that each point is excluded from its own neighbor search.
>
>
> **Q3: What type of KDE is used (i.e. kernel, bandwidth, etc.)?**
>
> For its simplicity and effectiveness, we use the `KDEMultivariate` function from the `statsmodel` library for density estimation. This function employs a Gaussian Kernel and applies the normal reference rule of thumb (i.e. bw=$1.06\hat{\sigma} n^{-1 / 5}$) based on the the standard deviation $\hat{\sigma}$ and sample size $n$ to select an appropriate bandwidth. While it is possible to use other density estimation kernels such as `Exponential Kernel` in `Scikit Learn`, we found that the Gaussian kernel coupled with the normal reference rule for bandwidth selection generally yields better performance across various models and datasets.
>
>
> **Q4: How are ECE and ACE computed (i.e. binning scheme)?**
>
> We follow [1] to implement Expected Calibration Error (ECE) and Adaptive Calibration Error (ACE):
> 1) Firstly, we divide samples into 15 bins and compute every bin’s average confidence and accuracy.
> 2) Next, we compute the absolute difference between each bin's average confidence and its corresponding accuracy.
> 3) The final calibration error is measured using the weighted difference (the fraction of samples in each bin as the weight).
>
> The key distinction between ECE and ACE lies in the binning scheme: ECE divides bins with **equal-confidence intervals** while ACE uses an adaptive scheme that spaces the bin intervals to contain an **equal number of samples** in each bin.
>
> [1]Nixon J, Dusenberry MW, Zhang L, Jerfel G, Tran D. Measuring Calibration in Deep Learning. InCVPR workshops 2019 Jun 16 (Vol. 2, No. 7).
>
> **Q5: Do you set aside calibration data for the scaling methods (TS, ETS, PTS, PTSK) in addition to the set aside data for ProCal?**
>
> In our implementation, we **do not reserve additional data** specifically for ProCal. Instead, we employ the same validation set used to train other calibration methods (e.g. TS). We designed it this way considering the typical constraints of validation set data availability and the cost associated with acquiring extra data. However, if sufficient validation set points are available, we do recommend setting aside separate calibration data for ProCal in case that other calibration methods might overfit their validation sets.

---

> > ### Comment · Reviewer_4gc8 · 2023-08-17
> >
> > These clarifications are useful, thank you (it would also be useful to include some of these in the main body of the revision) - I keep my score.

---

### Official Review · Reviewer_1D75 · 2023-07-06

**Soundness:** 4 excellent
**Presentation:** 3 good
**Contribution:** 3 good
**Rating:** 7
**Confidence:** 5

**Summary:**

The article focuses on the problem of uncertainty quantification in classification.
Calibration provides some guarantees on the estimated class probabilities on average. However, subgroups can still be miscalibrated. The article first aims to characterize these subgroup miscalibrations through proximity levels of the samples. It claims that a classifier, even calibrated, tends to be underconfident on high-proximity samples and overconfident on low-proximity samples. To measure this effect, it defines a proximity-informed ECE. Then, it proposes a recalibration framework based on this proximity-informed measure. Finally, it benchmarks the proposed method on numerous datasets and models.

**Strengths:**

* The problem is well presented and motivated. Figure 1 is pedagogical and helps the comprehension of the problem.
* The idea of characterizing subgroup miscalibrations through proximity is interesting, and refining uncertainty estimates is a good direction.
* The proposed framework is versatile: it can work as a stand-alone or combined with standard calibration techniques. It provides two versions: binning-based and continuous.
* The experimental study is substantial.
  * Datasets are large-scale, numerous, and multimodal: ImageNet, Yahoo-Topics, iNaturalist, ImageNet-LT, MultiNLI, ImageNet-C.
  * The article studies numerous models, e.g. 504 pre-trained models on ImageNet.
  * It compares many standard calibration methods, both scaling-based and histogram-based: temperature scaling, ensemble temperature scaling, parameterized temperature scaling, histogram binning, isotonic regression, and multi-isotonic regression.
* The experimental study provides substantial evidence.
  * It reveals proximity bias in most of the 504 pre-trained networks on ImageNet (72% according to a Wilcoxon rank-sum test).
  * The proposed method consistently improves over standard calibration methods.
  * The time overhead of the method is small, with an increase of 1.17% in inference runtime.
* Completeness of the study: It reveals the proximity bias, proposes a metric to measure it, a recalibration procedure to address it, and substantial experiments showing consistent improvements.

**Weaknesses:**

No major weaknesses.

**Questions:**

* In equation (5) on the confidence score adjustments, why a weight hyperparameter $\lambda \in (0, 1]$ is needed? Don't we want to take $\lambda=1$ to minimize PIECE? The paper talks quickly about regularization. What is the tradeoff? Are results wrong with $\lambda$ too close to 1? Do you take $\lambda=0.5$ everywhere in the experiments? How to choose it?
* Concerning the time overhead introduced by the recalibration methods, isn't it weird that Bin-Mean-Shift has only twice the overhead of isotonic regression (+0.1s vs +0.05s)? Since isotonic regression just needs to apply the mapping function from [0, 1] to [0, 1] to the inferred sample, while Bin-Mean-Shift needs to compute the distance from the new sample to all samples from the calibration set. The same question applies to mapping-based recalibration methods such as histogram binning (+0.03s).
* Could you develop when to use the histogram-based version (section 5.2) and when to use the continuous version (section 5.1) of your method?


---


Typos:
* L148: it should be 80% instead of 85%.
* L162: repeated word Appendix.

**Limitations:**

The authors discuss the following limitations:
* The proposed recalibration technique needs to maintain the calibration set during inference to compute the proximity of the new points.
* Maintaining the calibration set for inference may challenge the method when memory is limited.
* Focus limited to the closed-set multi-class classification problem.

---

> ### Author Rebuttal · Authors · 2023-08-09
>
> We sincerely thank reviewer 1D75 for the constructive comments and we are glad that the reviewer finds the problem interesting, the solution a good direction, and the study complete. Here we answer all the questions and hope they can address the concerns.
>
> **Q1: In equation (5) on the confidence score adjustments, why a weight hyperparameter \lambda is needed? Don't we want to take \lambda=1 to minimize PIECE? What is the tradeoff? Are results wrong with  too close to 1? Do you take \lambda=0.5 everywhere in the experiments? How to choose it?**
>
> To start, we revisit equation (5): $\hat{P}_{\text {ours }}=\hat{P}+\lambda \cdot\left(\mathcal{A}\left(B\_{m h}\right)-\mathcal{F}\left(B\_{m h}\right)\right)$.
>
> First, we agree with you that setting $\lambda=1$ in Equation (5) would ideally calibrate the model and minimize PIECE more effectively. However, in practice, we often encounter bins with a smaller number of samples, whose estimate will have high variance. In such cases, the calculated $F(B_{mh}) -A(B_{mh})$ can be quite inaccurate as the estimate of the corresponding population quantity. To reduce variance in these scenarios, we introduce a **shrinkage coefficient, $\lambda$**. By setting a smaller $\lambda$, we can reduce the variance and make the final prediction more accurate at the expense of biased estimator.
>
> Therefore, the use of $\lambda$ can be seen as a form of bias-variance trade-off. By adjusting the value of $\lambda$, we are able to control the balance between bias and variance in the model's output. In practice, we choose $\lambda=0.5$ as a reasonable default for all our experiments, which we find offers consistent performance across various settings.
>
> **Q2: Why Bin-Mean-Shift has only twice the overhead of isotonic regression (+0.1s vs +0.05s)? Isotonic regression just needs to apply the mapping function from [0, 1] to [0, 1] to the inferred sample, while Bin-Mean-Shift needs to compute the distance from the new sample to all samples from the calibration set. The same question applies to mapping-based recalibration methods such as histogram binning (+0.03s).**
>
> We attribute our efficiency to the recent advances in neighborhood search algorithms.
> In our paper, we employ `indexFlatL2` from the Meta open-sourced GPU-accelerated Faiss library[2] to calculate each sample's nearest neighbor. This algorithm enables us to **reduce the time for nearest neighbor search to approximately 0.04 ms per sample** (shown below). The computation overhead beyond the neighbor search is actually quite similar to isotonic regression, which leads to the total time being roughly twice that of isotonic regression **(0.04 + 0.05 ≈ 0.1s)**.
>
> For comparison, here's the inference time comparison (ms per sample) for our method and baseline methods on ImageNet (using a ViT/B-16@224px model on a single Nvidia GTX 2080 Ti, with all settings following those in our paper, and averaged across 10 runs):
> |                  |  NS   |Conf  | TS | HB | IR | PTS | PTSK  |ETS| MIR| BIN* | DEN* |
> |:---------------------:|:-----:|:---------:|:---------------------------:|:---------:|:---------:|:---------:|:---------:|:---------:|:---------:|:---------:|:---------:|
> |  Time(ms) |0.04   |5.60|  5.60 | 5.63| 5.65 | 5.61 |5.66 |5.64|  5.84  | 5.70 | 6.29|
>
> NS represents the time consumption for **neighbor search**. BIN* and DEN* are our proposed methods, where BIN denotes Bin Mean-Shift for discrete output and DEN is Density-Ratio Calibration for continuous output.
>
> **Q3: Could you develop when to use the histogram-based version (section 5.2) and when to use the continuous version (section 5.1) of your method?**
>
> Thanks for your constructive question. First, we would like to clarify that existing calibration algorithms can be classified into 2 types:
> 1) those producing **continuous outputs** such as softmax, temperature scaling (TS), and other methods (ETS, PTS, PTSK) which scale the input scores in a continuous way; and
> 2) those producing **discrete outputs**, particularly binning or isotonic regression methods (HB, IR, MIR) which group the samples and output the exact same scores for samples in the same group.
>
> Our density-ratio calibration works by estimating continuous density functions, and is suited for the 1st case when the input confidence scores are continuous. In contrast, our bin-mean-shift method does not rely on densities and is more suited for utilizing outputs from the 2nd category of methods, i.e. discrete outputs. In combination, our 2 calibration approaches form a plug-and-play framework that is applicable to confidence scores of either type. We will revise the paper to clarify this.

---

> > ### Comment · Reviewer_1D75 · 2023-08-17
> >
> > I thank the authors for their detailed rebuttal.
> >
> > Concerning the answer to Q1 on the choice of the $\lambda$ coefficient, according to the authors, it appears that $\lambda = 1$ can be problematic when there are too few samples in the bins. I wonder if this work could benefit from the derivations of [31] section C.5. They face the same problem when estimating the inter-region variance when too few samples are in the regions (see the $GL_{plugin}$ curve in Fig. 1.a. which is overestimated when the number of samples per region is low, versus the $GL_{LB}$ curve, i.e., the $GL_{plugin}$ curve corrected with debiasing, which is correctly estimated even when the number of samples per region is low). I would guess that a debiasing similar to the one of section C.5 [31] could be applied to this work, which could enable values of $\lambda$ closer or equal to 1, even when a few samples are in the bins.
> >
> > Answer to Q2: the computational efficiency of the neighborhood search is quite impressive.
> >
> > Answer to Q3: Thanks for clarifying and revising the paper.
> >
> >
> > [31] Alexandre Perez-Lebel, Marine Le Morvan, and Gaël Varoquaux. Beyond calibration: estimating the grouping loss of modern neural networks. In ICLR, 2023.

---

### Official Review · Reviewer_Wiui · 2023-07-26

**Soundness:** 3 good
**Presentation:** 3 good
**Contribution:** 3 good
**Rating:** 6
**Confidence:** 3

**Summary:**

This work addresses the problem of proximity bias and confidence calibration by performing a comprehensive empirical study of various pretrained ImageNet models. The empirical findings provide insights on persistence of proximity bias even after performing calibration using existing post-hoc calibration algorithms. To mitiagte proximity bias and improve confidence calibration based on sample proximity, the paper proposes PROCAL algorithm that can be used as a plug-and-play method combined with existing calibration approaches. Further, proximity-informed expected calibration error metric is introduced to quantify the effectiveness of calibration algorithms in mitigating proximity bias.

**Strengths:**

- A comprehensive study of pretrained ImageNet models involving various neural network architectures on their model calibration and proximity bias evaluation. The empirical study is performed on image classification tasks (under balanced, long-tail, and distribution-shift settings) and text classification tasks.
- Experimental evaluation if thorough using various calibration metrics.
- The paper provides many interesting observations from the empirical study related to proximity bias and model calibration, which is an important area of study under long-tailed data distribution settings.

**Weaknesses:**

- The presentation of experimental results and writeup of Experiments Section 6 can be improvised, the findings for many of the questions are pointed to the Appendix without any brief details in the manuscript. I understand this is due to page limitation, but I would suggest the authors to focus on the results that are presented in main manuscript, or include the brief info in the manuscript at least.


**Questions:**

- Are the iNaturalist 2021 results in Table 1 based on transfer learning from pretrained ImageNet models, or directly evaluated on pretrained backbone?
- What are the criteria to choose between density-ratio calibration versus Bin-Mean-shift approach?
- typo in line#309 ImageNet-TL --> ImageNet-LT

**Limitations:**

The authors have addressed the limitations of their work in the manuscript.

---

> ### Author Rebuttal · Authors · 2023-08-08
>
> We sincerely thank reviewer Wiui for the constructive suggestions on the writing of the experiment part. We are also glad that the reviewer finds the phenomenon study comprehensive, the experiment evaluation thorough, the observation interesting, and the issue we identify to be important. Here we answer all the questions and hope they can address the concerns.
>
> **Q1: Are the iNaturalist 2021 results in Table 1 based on transfer learning from pretrained ImageNet models, or directly evaluated on pretrained backbone?**
>
> The results are directly evaluated on the **pretrained backbone** which we follow this paper[2] and download from the repo: https://github.com/visipedia/newt/blob/main/benchmark/README.md
>
> [2] Van Horn G, Cole E, Beery S, Wilber K, Belongie S, Mac Aodha O. Benchmarking representation learning for natural world image collections. CVPR 2021.
>
>
> **Q2: What are the criteria to choose between density-ratio calibration versus Bin-Mean-shift approach?**
>
> Thanks for your constructive question. First, we would like to clarify that existing calibration algorithms can be classified into 2 types:
>
> 1) those producing *continuous outputs* such as softmax, temperature scaling (TS), and other methods (ETS, PTS, PTSK) which scale the input scores in a continuous way; and
> 2) those producing *discrete outputs*, particularly binning or isotonic regression methods (HB, IR, MIR) which group the samples and output the exact same scores for samples in the same group.
>
> Our density-ratio calibration works by estimating continuous density functions, and is suited for the 1st case when the input confidence scores are continuous. In contrast, our bin-mean-shift method does not rely on densities and is more suited for utilizing outputs from the 2nd category of methods, i.e. discrete outputs. In combination, our 2 calibration approaches form a plug-and-play framework that is applicable to confidence scores of either type. We will revise the paper to clarify this.
>
> **W1: The presentation of experimental results and writeup of Experiments Section 6 can be improved, the findings for many of the questions are pointed to the Appendix without any brief details in the manuscript. I understand this is due to page limitation, but I would suggest the authors focus on the results that are presented in the main manuscript, or include the brief info in the manuscript at least.**
>
> We appreciate your suggestions, and we will include a brief summary of the results when making reference to the appendix to improve the readability and flow of the paper in the future versions.

---

> > ### Comment · Reviewer_Wiui · 2023-08-17
> >
> > I thank the authors for the rebuttal. I had read the responses and I retain my score.

---

### Author Rebuttal · Authors · 2023-08-10

**The following questions primarily pertain to the details of how to compute the bias index, and therefore, we have grouped them together for convenience and clarity.**

To provide further clarity, let's revisit the process of how we calculate the bias index:
1. To do the hypothesis testing, we first split the samples into 5 equal-sized proximity groups and select the highest and lowest proximity groups.
2. From the high proximity group, we randomly select 10,000 points and find corresponding points in the low proximity group that have similar confidence levels.
3. Next, we reverse this process to randomly select 10,000 points from the low proximity group and find corresponding points in the high proximity group with matched confidence.
4. We therefore merge all the points from the high proximity group into $B_H$ and those from the low proximity group into $B_L$, with the $B_H$ and $B_L$ having similar average confidence.
5. We apply the Wilcoxon rank-sum test to evaluate whether the mean difference in their accuracy between $B_H$ and $B_L$ is significantly different from zero.

**Q4.1: what is the sensitivity of two points having the 'same' confidence?**

If we understand correctly, the reviewer is asking whether we can ensure that two points have the same confidence. Firstly, we would like to clarify that *the two samples have the **same confidence*** actually means they have the *closest confidence in the validation set with the maximum difference less than 0.05*. In practice, we apply nearest neighbor search to find points in the alternate group with approximately the same confidence (i.e. closest possible confidence) to our target. Pairs with significant confidence difference (> 0.05) are excluded, ensuring that $B_H$ and $B_L$ have comparable confidence levels.

**Q4.2:  Is there a large difference of average confidence between the two, and how well do their confidences intersect?**

If we understand correctly, 'the two' refers to the original groups with the highest and lowest proximities in step 1).
1. In this case, there is indeed an observable difference in their average confidence, with high proximity samples having higher average confidence. Therefore, we apply nearest neighbor search and reject policy to ensure the sampled $B_H$ and $B_L$ in step 5 have almost the same confidence.
2. As we only divide samples into 5 equally-sized groups, their confidences intersect well. When implementing this algorithm, users can also visualize the confidence distribution to verify whether overlaps occur. If their confidence levels have no overlap, we suggest reducing the number of splits from 5 to 3 to ensure low/high proximity groups have similar confidence but different proximities.

**Q4.3: How does the Bias Index changed if only B_H is used for sampling points and B_L is used for matching, and vice versa?**

Thank you for your question. As suggested in step 2 and 3, we do not just use the $B_H$ for sampling and $B_L$ for matching; both groups are used for sampling and matching.
In addition, as we highlighted in Q4.1, we employ the nearest neighbor search and reject strategy to ensure that $B_H$ and $B_L$ have similar average confidences. This means that the samples drawn primarily fall within the range of intersecting confidences, resulting in minimal changes to the bias index even if we randomly choose one group for sampling.

**Q4.4: What if the intermediate groups are used?**

We also conducted experiments in which we divided samples into 3 equal-sized proximity groups, which reduced the proximity difference between the high/low groups (can be seen as the proxy of intermediate groups). We found that the same observations hold true in this context.

---

### Decision · Program_Chairs · 2023-09-21

**Decision:**

Accept (spotlight)

**Comment:**

The authors examine the phenomenon of miscalibration in neural network classifiers.  They find empirically that typical ImageNet trained classifiers have significantly different levels of calibration across "low-proximity" and "high-proximity" subpopulations, where they define higher-proximity samples as those close to many other samples while low-proximity samples are far from other samples (where the distance is measured using the Euclidean distance of the features from the penultimate layer of the neural net).  They show that the standard approach for rectifying miscalibration, temperature scaling, does not remove the proximity bias of the classifiers.  On the other hand, they propose a new method, PIECE, which does remove this bias.

The reviewers were unanimous in their support for the submission.  They were impressed by the empirical findings on the existence of proximity bias as well as the proposed method to address this bias.  I concur with the reviewers and I recommend this paper for acceptance with spotlight.